# Unifying over-smoothing and over-squashing in graph neural networks: A physics informed approach and beyond

## Abstract

Graph Neural Networks (GNNs) have emerged as one of the leading approaches for machine learning on graph-structured data. Despite their great success, critical computational challenges such as over-smoothing, over-squashing, and limited expressive power continue to impact the performance of GNNs. In this study, inspired from the time-reversal principle commonly utilized in classical and quantum physics, we reverse the time direction of the graph heat equation. The resulted reversing process yields a class of high pass filtering functions that enhance the sharpness of graph node features. Leveraging this concept, we introduce the Multi-Scaled Heat Kernel based GNN (MHKG) by amalgamating diverse filtering functions' effects on node features. To explore more flexible filtering conditions, we further generalize MHKG into a model termed G-MHKG and thoroughly show the roles of each element in controlling over-smoothing, over-squashing and expressive power. Notably, we illustrate that all aforementioned issues can be characterized and analyzed via the properties of the filtering functions, and uncover a trade-off between over-smoothing and over-squashing: enhancing node feature sharpness will make model suffer more from over-squashing, and vice versa. Furthermore, we manipulate the time again to show how G-MHKG can handle both two issues under mild conditions. Our conclusive experiments highlight the effectiveness of proposed models. It surpasses several GNN baseline models in performance across graph datasets characterized by both homophily and heterophily.

## 1 Introduction

Graph Neural Networks (GNNs) have demonstrated exceptional performance in learning the representations of graph-structured data Ji et al. (2021); Wu et al. (2020). Within the diverse research avenues of GNNs, one prominent aspect is to interpret GNNs via different perspectives, such as gradient flow Di Giovanni et al. (2022); Han et al. (2022), neural diffusion Thorpe et al. (2021); Chamberlain et al. (2021b), and testing of graph isomorphism Xu et al. (2019), all of which contribute to a deeper comprehension of GNNs' underlying working mechanism. Concurrently, another trajectory in GNN's development is oriented toward addressing specific challenges. Generally, three inherent issues in GNNs have been identified: over-smoothing Cai & Wang (2020), over-squashing Topping et al. (2021), and limitations in expressive power Xu et al. (2019), each of them is openly acknowledged and arises due to various aspects of GNNs and its graph data input. As such, research efforts are primarily focused on tackling these aspects to effectively address the mentioned problems. For instance, to counteract the over-smoothing phenomenon, researchers have advocated for the augmentation of feature variation by incorporating a source term Thorpe et al. (2021) or modulating the energy regularization to prevent rapid diminishment Shao et al. (2022); Fu et al. (2022). To mitigate over-squashing, strategies may encompass exploring graph topology through methods such as *graph re-weighting* Shi et al. (2023a) and *graph rewiring* Topping et al. (2021). Lastly, enhancing the capacity to discern non-isomorphic graphs could be achieved by assigning more intricate embeddings to graph nodes Xu et al. (2019); Wijesinghe & Wang (2021).

While many efforts have been made on these three issues, little research (Nguyen et al., 2023; Giraldo et al., 2022) have been done to consider these issues in a unified manner to reveal their underlying

relationships and further seeking for mitigating them via one specific GNN and an unified way. The main difficulties for this situation may be due to

- Some GNNs lack the necessary flexibility to serve as the subjects for investigating the aforementioned triad of issues in a cohesive manner.
- The pursuit of greater flexibility in GNNs in general leads to increased model complexity. Consequently, effectively exploring the roles on GNNs components on these issues becomes a formidable task.
- There's a gap in research regarding the transition between addressing over-smoothing, typically focusing on node features, and tackling over-squashing and enhancing expressive power, which often concentrate on graph topology.

In the present study, we derive inspiration from the physical reality of graph heat equation, leading to a novel perspective that examines the propagation of node features in the reverse direction of time. We demonstrate that many prevailing GNNs can invert the feature propagation process, transitioning from smoothing to sharpening effects, and vice versa. Accordingly, we propose a Multi-Scale Heat Kernel GNN (MHKG) that propagates node features via the balance between smoothing and sharpening effects induced by the low and high pass spectral filtering functions generated from the heat and reverse heat kernels. Furthermore, we generalize the MHKG into a more flexible model, referred to as G-MHKG, and provide a comprehensive examination of the components within G-MHKG that regulate over-smoothing, over-squashing and expressive power. Notably, by utilizing G-MHKG as an analytical tool and examining the properties of the associated filtering functions, we uncover a trade-off between over-squashing and over-smoothing in the graph spectral domain. This relationship is revealed under mild conditions on the filtering functions, offering fundamental insights between these issues. Lastly, we manipulate the time again in G-MHKG to show that its capability of sufficiently handling both issues for heterophily graphs and it is impossible to achieve the same goal for homophily graphs.

**Contribution and Outline** Our main goal is to illustrate and verify the underlying relationship between aforementioned issues via our proposed model inspired by the time reversal principle from physics. In addition, we aim to show how our proposed model can handle these issues in an unified manner. In Section 3, we explore the link between graph filtering functions and solutions of a class of ordinary differential equations (ODEs) on graph node features. This connection propels the introduction of two multi-scale heat kernel based models: MHKG and the generalized MHKG (G-MHKG). In Section 4 we show how the filtering matrices in our model control the energy dynamics of node features, thereby effectively addressing the over-smoothing challenge. In Section 5, we demonstrate the weight matrix in G-MHKG determines the model's expressive power and over-squashing. More importantly, in Section 6 and 7 we show there is trade-off between over-smoothing and over-squashing. We also prove that it is impossible to sufficiently handle both two issues for homophily graphs and for heterophily graphs G-MHKG can handle these issues by a simple manipulation of time. We verify our theoretical claims via empirical studies in Section 8.

## 2 Preliminaries

**Graph basics and GNNs** To begin, we let $\mathcal{G} = (\mathcal{V}, \mathcal{E})$ with nodes set $\mathcal{V} = \{v_1, v_2, \cdots, v_N\}$ of total $N$ nodes and edge set $\mathcal{E} \subseteq \mathcal{V} \times \mathcal{V}$. We also denote the graph adjacency matrix as $\mathbf{A} \in \mathbb{R}^{N \times N}$. In addition, let $\widetilde{\mathbf{A}} = \mathbf{A} + \mathbf{I}_N$ be the adjacency matrix with self-loop on each node, one can define the normalized adjacency matrix as $\widehat{\mathbf{A}} = \mathbf{D}^{-1/2} \widetilde{\mathbf{A}} \mathbf{D}^{-1/2}$ where $\mathbf{D}$ is the diagonal degree matrix, with the $i$-th diagonal entry given by $d_i = \sum_j \widetilde{a}_{ij}$ the degree of node $i$. The normalized graph Laplacian is given by $\widehat{\mathbf{L}} = \mathbf{I} - \widehat{\mathbf{A}}$. We further let $\rho_{\widehat{\mathbf{L}}}$ denote the largest eigenvalue (also called the highest frequency) of $\widehat{\mathbf{L}}$. In terms of GNNs, we note that in general there are two types of GNNs, the spatial-based GNNs such as graph convolution network (GCN) (Kipf & Welling, 2016) defines the layer-wise propagation rule via the normalized adjacency matrix as

$$\mathbf{H}^{(t)} = \sigma\big(\widehat{\mathbf{A}} \mathbf{H}^{(t-1)} \mathbf{W}^{(t)}\big), \tag{1}$$

where we let $\mathbf{H}^{(t)}$ as the feature matrix at layer $t$ with $\mathbf{H}^{(0)} = \mathbf{X} \in \mathbb{R}^{N \times c}$, the input feature matrix, and $\mathbf{W}^{(t)}$ is the learnable weight matrix performing channel mixing. On the other hand, spectral

GNNs such as ChebyNet (Defferrard et al., 2016) perform spectral filtering on the spectral domain of the graph as

$$\mathbf{H}^{(t)} = \sigma\left(\mathbf{U}g_\theta(\boldsymbol{\Lambda})\mathbf{U}^\top\mathbf{H}^{(0)}\right), \tag{2}$$

where $g_\theta(\boldsymbol{\Lambda})$ serves as the filtering function on normalized Laplacian, which utilizes an eigendecomposition $\widehat{\mathbf{L}} = \mathbf{U}\boldsymbol{\Lambda}\mathbf{U}^\top$ and $\mathbf{U}^\top\mathbf{h}$ is known as the Fourier transform of a graph signal $\mathbf{h} \in \mathbb{R}^N$. In this paper, we let $\{(\lambda_i, \mathbf{u}_i)\}_{i=1}^N$ be the set of eigenvalue and eigenvector pairs of $\widehat{\mathbf{L}}$ where $\mathbf{u}_i$ are the column vectors of $\mathbf{U}$.

**Graph heat kernel** Generalizing from the so-called heat equation defined on a manifold, one can define graph heat equation as $\frac{\partial \mathbf{H}^{(t)}}{\partial t} = -\widehat{\mathbf{L}}\mathbf{H}^{(t)}$, in which $\mathbf{H}^{(t)}$ is the feature representation at a specific iteration time $t$. The solution of the graph heat equation, denoted as $\mathbf{H}^{(t)}$, is given by $\mathbf{H}^{(t)} = e^{-t\widehat{\mathbf{L}}}\mathbf{H}^{(0)}$ with $\mathbf{H}^{(0)} = \mathbf{X}$ as the initial condition. It is well-known that the application of Euler discretization leads to the propagation of the linear GCN models (Kipf & Welling, 2016; Wu et al., 2019) and this process is with the name of Laplacian smoothing (Chung, 1997). The characteristic of the solution is indeed governed by the so-called Heat Kernel, denoted by $\mathbf{K}_t = e^{-t\widehat{\mathbf{L}}}$. The heat kernel defines a continuous-time random walk, and defines a semi-group, i.e., $\mathbf{H}^{(t+s)} = \mathbf{H}^{(t)} \cdot \mathbf{H}^{(s)}$ for any $t, s \geq 0$, and $\lim_{t\to\infty} \mathbf{H}^{(t)} = \mathbf{I}$ (Chung, 1997). This fact indicates non-distinguishment for the nodes with same degrees, known as over-smoothing. We included a more detailed discussion on both heat operator and heat kernel in Appendix A.1

## 3 TURNING SMOOTHING TO SHARPENING

### 3.1 KERNEL & FILTER CORRESPONDENCE: A PHYSICS INFORMED APPROACH

We now delve deeper into the graph heat kernel $\mathbf{K}_t$. At a specific time $t$, if one considers $t$ is flowed at integer interval, then $\mathbf{H}^{(t)} = e^{-t\widehat{\mathbf{L}}}\mathbf{H}^{(0)}$ can be interpreted as a linear diffusion (i.e., $e^{-\widehat{\mathbf{L}}}$) on $\mathbf{H}^{(0)}$ repeated for $t$ times. Consequently, at each step, one can treat $e^{-\widehat{\mathbf{L}}}$ as a low pass filtering function (*monotonically decreasing*) on the graph spectra. More generally, one can assign one function $f$ onto $\widehat{\mathbf{L}}$ to build a class of ODEs with the form of $\frac{\partial \mathbf{H}^{(t)}}{\partial t} = -f(\widehat{\mathbf{L}})\mathbf{H}^{(t)}$ and, similar to the basic heat equation, we can consider the generalized heat kernel $\widehat{\mathbf{K}}_t = e^{-tf(\widehat{\mathbf{L}})}$. We note that in the sequel, $f(\widehat{\mathbf{L}})$ serve as a filtering function (i.e., polynomial or analytic functions) acting element-wisely on the eigenvalues of $\widehat{\mathbf{L}}$, that is $f(\widehat{\mathbf{L}}) = \mathbf{U}f(\boldsymbol{\Lambda})\mathbf{U}^\top$.

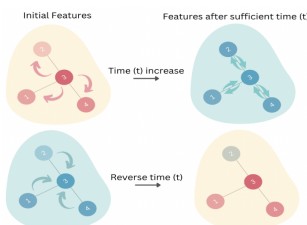

Figure 1: Top: The evolution of the node feature of diffusion (smoothing) process (i.e., from distinct features to over-smoothing). Bottom: the reverse diffusion (sharpening) process (i.e., from nearly identical to distinct node features).

A physical interpretation of the aforementioned class of ODEs suggests that heat flows in a constant direction and speed. This aligns with the *second law of thermodynamics*, indicating an inevitable evolution of the heat distribution in an isolated system, leading to what is known as *thermal equilibrium*. In analogy to GNNs, such phenomenon indicates all node features eventually become equal to each other, typically known as the over-smoothing issue. Together with the recent challenge on fitting GNNs to the so-called heterophily graphs, GNNs are preferred to produce a mixed dynamic, involving not only smoothing but also sharpening the features of connected nodes with different labels. Therefore, **we wish to turn the diffusion into a reverse direction** which is equivalent to providing external (anti-)force to the system and expect to achieve a fusion of heat in certain areas of the system while allowing diffusion in others during each evolution step. Since for any given $f$ on $\widehat{\mathbf{L}}$, one can induce another system that produces reverse process by assigning a negative sign before $f$. Accordingly, one can obtain a reverse filtering effect once the diffusion direction is turned. Additionally, if we further require the filtering process $\mathcal{F} : \mathbb{R}^{N\times c} \to \mathbb{R}^{N\times c}$ such that $\mathcal{F}(\mathbf{H}) = e^{-tf(\widehat{\mathbf{L}})}\mathbf{H}^{(0)}$ to be bijective[1], then by replacing $t$ to the $-t$, one can recover $\mathbf{H}^{(0)}$ from

---

[1]Other requirements such as homeomorphism or isomorphism can also be applied.

$\mathbf{H}^{(t)}$. We note that, in this case, the propagation and recovery of $\mathbf{H}^{(0)}$ is aligned with those physical variables (i.e., electric potential, energy density of the electromagnetic field) that are unchanged through inverse time operators in both classic and quantum mechanics. Figure 1 shows how graph diffusion smooths the node feature and reverse diffusion makes features more distinct. With this understanding, we next show how the novel heat kernel GCN is constructed.

## 3.2 MULTI-SCALE HEAT KERNEL GCN (MHKG)

Followed by the idea that at each discrete time step, i.e., from $\ell - 1$ to $\ell$, the heat kernel based GCN model shall possess mixed dynamic (smoothing & sharpening) on node features. We thus propose the multi-scale heat kernel GCN (MHKG) by combining the two directions of heat flow governed by heat kernel $e^{-f(\widehat{\mathbf{L}})}$ and reverse heat kernel $e^{f(\widehat{\mathbf{L}})}$:

$$\mathbf{H}^{(\ell)} = \mathbf{U}\mathrm{diag}(\theta_1)\mathbf{\Lambda}_1\mathbf{U}^\top\mathbf{H}^{(\ell-1)}\mathbf{W}^{(\ell-1)} + \mathbf{U}\mathrm{diag}(\theta_2)\mathbf{\Lambda}_2\mathbf{U}^\top\mathbf{H}^{(\ell-1)}\mathbf{W}^{(\ell-1)}, \quad (3)$$

in which $\mathrm{diag}(\theta_1)$ and $\mathrm{diag}(\theta_2)$ are learnable filtering matrices with $\mathbf{\Lambda}_1 = -f(\mathbf{\Lambda}) = \mathrm{diag}(\{e^{-f(\lambda_i)}\}_{i=1}^N)$ and $\mathbf{\Lambda}_2 = f(\mathbf{\Lambda}) = \mathrm{diag}(\{e^{f(\lambda_i)}\}_{i=1}^N)$. Referring to the previous note on $f$, one can denote $f_\theta(\widehat{\mathbf{L}}) = \mathbf{U}\mathrm{diag}(\theta)f(\mathbf{\Lambda})\mathbf{U}^\top$.

**Beyond time reversal: an even more general case**    In light of the construction of MHKG, one can consider a more generalized model with more flexible choice of dynamics as:

$$\frac{\partial\mathbf{H}^{(t)}}{\partial t} = f(\widehat{\mathbf{L}})\mathbf{H}^{(t)}, \quad \frac{\partial\mathbf{H}^{(t)}}{\partial t} = g(\widehat{\mathbf{L}})\mathbf{H}^{(t)}, \quad (4)$$

where unlike in (3), $g(\widehat{\mathbf{L}})$ not necessarily equals $-f(\widehat{\mathbf{L}})$. For simplicity, in this paper we only consider two fixed dynamic on $\mathbf{H}$ and the conclusion we provided can be easily generalized to the multiple dynamic cases. The corresponding GNN induced from Eq. (4) is:

$$\mathbf{H}^{(\ell)} = \mathbf{U}\mathrm{diag}(\theta_1)e^{f(\mathbf{\Lambda})}\mathbf{U}^\top\mathbf{H}^{(\ell-1)}\mathbf{W}^{(\ell-1)} + \mathbf{U}\mathrm{diag}(\theta_2)e^{g(\mathbf{\Lambda})}\mathbf{U}^\top\mathbf{H}^{(\ell-1)}\mathbf{W}^{(\ell-1)}. \quad (5)$$

We named this generalized model as G-MHKG. It is not difficult to verify that G-MHKG is a general form of various GNN models. The simplest case one can obtain is to set $\theta_2 = \mathbf{0}_N$ and $e^{f(\mathbf{\Lambda})} = e^{-\xi\mathbf{\Lambda}}$, where $\xi \in \mathbb{R}$ is a constant or learnable coefficient, we recover the GraphHeat proposed in (Zhao et al., 2020). If we let $g(\mathbf{\Lambda}) = \mathbf{0}_{N \times N}$, and $\theta_2 = \mathbf{c}_N$, where $\mathbf{c}_N$ as an $N$-dimensional vector with arbitrary constant as $c$, then the model is equivalent to GRAND++ (Thorpe et al., 2022) with a layer dependent source term, see Appendix B for more detailed illustrations. In the following sections, we show the roles of $\theta$ and $\mathbf{W}$ in controlling the three aforementioned issues of GNNs.

## 4 FILTERING MATRICES CONTROL MODEL DYNAMICS & OVER-SMOOTHING

In this section, we illustrate the role of the filtering matrices, i.e., $\mathrm{diag}(\theta_1)$ and $\mathrm{diag}(\theta_2)$, in G-MHKG control model's energy dynamic and how a controllable energy dynamic affects model's adaption on homophily and heterophily graphs. For the simplicity of the analysis we further assume that $f(\cdot)$ and $g(\cdot)$ to be *monotonically decrease/increase* in the spectral domain of the graph so that they can be considered as low/high pass filtering functions. We note that this assumption is aligned with the settings in many recent works such as (Zheng et al., 2022; Liu et al., 2023; Lin & Gao, 2023) and the conclusion we represent can be easily applied to MHKG. To quantify model's energy dynamics, we consider the Dirichlet energy of the node features $\mathbf{H}$, defined by $\mathbf{E}(\mathbf{H}) = \mathrm{Tr}(\mathbf{H}^\top\widehat{\mathbf{L}}\mathbf{H})$. It is well-known that Dirichlet energy becomes 0 when the model encounters over-smoothing issue. To represent such asymptotic energy behavior, Di Giovanni et al. (2022); Han et al. (2022); Shi et al. (2023b) consider a general dynamic as $\dot{\mathbf{H}}^{(t)} = GNN_\theta(\mathbf{H}^{(t)}, t)$, with $GNN_\theta(\cdot)$ as an arbitrary GNN function characterising its behavior by low/high-frequency-dominance (L/HFD).

**Definition 1** ((Di Giovanni et al., 2022))**.** $\dot{\mathbf{H}}^{(t)} = GNN_\theta(\mathbf{H}^{(t)}, t)$ is Low-Frequency-Dominant (LFD) if $\mathbf{E}(\mathbf{H}^{(t)}/\|\mathbf{H}^{(t)}\|) \to 0$ as $t \to \infty$, and is High-Frequency-Dominant (HFD) if $\mathbf{E}(\mathbf{H}^{(t)}/\|\mathbf{H}^{(t)}\|) \to \rho_{\widehat{\mathbf{L}}}/2$ as $t \to \infty$, where $\rho_{\widehat{\mathbf{L}}}$ stands for the largest eigenvalue of $\widehat{\mathbf{L}}$.

**Lemma 1** ((Di Giovanni et al., 2022))**.** *A GNN model is LFD (resp. HFD) if and only if for each* $t_j \to \infty$*, there exists a sub-sequence indexed by* $t_{j_\kappa} \to \infty$ *and* $\mathbf{H}_\infty$ *such that* $\mathbf{H}_{j_\kappa}^{(t)}/\|\mathbf{H}_{j_\kappa}^{(t)}\| \to \mathbf{H}_\infty$ *and* $\widehat{\mathbf{L}}\mathbf{H}_\infty = 0$ *(resp.* $\widehat{\mathbf{L}}\mathbf{H}_\infty = \rho_{\widehat{\mathbf{L}}}\mathbf{H}_\infty$*).*

**Remark 1** (Dirichlet energy, graph homophily and heterophily). It has been shown (Bronstein et al., 2021) that if the graph is heterophily where the connected node are unlikely to share the same labels, one may prefer a GNN with sharpening effect, corresponding to increase of energy. Whereas, when the graph is highly homophily, a smoothing effect is preferred. Based on the settings above, we show our conclusion on the property of energy dynamic of G-MHKG in the following.

**Theorem 1.** *G-MHKG can induce both LFD and HFD dynamics. Specifically, let $\theta_1 = \mathbf{1}_N$ and $\theta_2 = \zeta \mathbf{1}_N$ with a positive constant $\zeta$. Then, with sufficient large $\zeta$ ($\zeta > 1$) so that $\mathrm{e}^{f(\mathbf{\Lambda})} + \zeta \mathrm{e}^{g(\mathbf{\Lambda})}$ is monotonically increasing on spectral domain, G-MHKG is HFD. Similarly, if $0 < \zeta < 1$ and is sufficient small such that $\mathrm{e}^{f(\mathbf{\Lambda})} + \zeta \mathrm{e}^{g(\mathbf{\Lambda})}$ is monotonically decreasing, the model is LFD.*

We leave the proof in Appendix C.1. Theorem 1 directly shows the benefits of constructing multi-scale GNNs because such a model provides flexible control of the dominant dynamic that always amplifies/shrinks the node feature differences at every step of its propagation. Accordingly, once the model is HFD, there will be no over-smoothing issue. Furthermore, it is straightforward to verify that single-scale GNN (Kipf & Welling, 2016; Chamberlain et al., 2021a) may satisfy the conditions of frequency dominance described in Theorem 1, and in fact some of them can never be HFD to fit heterophily graphs. See (Di Giovanni et al., 2022) for more details. Finally, we note that there are many other settings of $\theta_1$ and $\theta_2$ that make the model HFD, the example in Theorem 1 is one of them to illustrate its existence.

**Remark 2.** Recent research on GNNs dynamics (Di Giovanni et al., 2022) suggests that to induce HFD, the weight matrix $\mathbf{W}$ must be symmetric and contain at least one negative eigenvalue. However, from Theorem 1, G-MHKG can be L/HFD without considering the impact of $\mathbf{W}$. This further supports the flexibility and generalizability of G-MHKG. However, as we will illustrate in the next section, $\mathbf{W}$ has direct impact on the model's expressive power and over-squashing.

## 5 WEIGHTS CONTROL EXPRESSIVE POWER & OVER-SQUASHING

We have demonstrated in Section 4 that the L/HFD induced by G-MHKG does not impose any specific conditions on the weight matrix $\mathbf{W}$. However, it is intriguing to investigate the functionality of $\mathbf{W}$ in G-MHKG. Recent developments in (Di Giovanni et al., 2023) shed light on the fact that $\mathbf{W}$ significantly influences the model's expressive power which refers to the class of functions that GNN can learn, taking node features into consideration. This influence becomes evident through the pairwise (over) squashing phenomenon, which interacts with the number of layers denoted as $\ell$, i.e. the depth of the model. Specifically, let $\mathcal{Q} : \mathbb{R}^{N \times c} \to \mathbb{R}^{N \times d}$ be the function that GNNs aim to learn, where $d$ is the dimension of the node features after certain layers of propagation. One can characterize the **expressive power** of one GNN model by estimating the amount of mixing of $\mathcal{Q}(\mathbf{X})$ among any pair of nodes by the following definition.

**Definition 2** (Maximal Mixing). For a smooth function $\mathcal{Q}$ of $N \times c$ variables, the maximal mixing induced from $\mathcal{Q}$ associated with nodes $u, v$ can be measured as:

$$\mathrm{mix}_{\mathcal{Q}}(v, u) = \max_{\mathbf{X}} \left\| \frac{\partial(\mathcal{Q}(\mathbf{X}))_v}{\partial \mathbf{x}_u} \right\|, \tag{6}$$

where $\| \cdot \|$ is the spectral norm of the matrix. Here the $\mathrm{mix}_{\mathcal{Q}}(u, v) \in \mathbb{R}$ depends on the maximal component of Jacobian matrix between the output of GNN and the initial node features. Furthermore if one replaces the $\mathcal{Q}(\mathbf{X})$ in Eq. (6) by the feature representation from an $\ell$-layer GNN, the quantity presented in Eq. (6) aligns with the so-called *sensitivity* which was proposed to measure the over-squashing issue (Topping et al., 2021). Therefore, it is natural to see that if **one GNN has strong mixing (expressive) power, the over-squashing issue in this GNN tends to be smaller**. We note that similar definitions and discussions can be found in (Di Giovanni et al., 2023). Accordingly, we define **over-squashing** (OSQ) as

$$\mathrm{OSQ}_{v,u} = \left( \mathrm{mix}_{\mathcal{Q}}(v, u) \right)^{-1}. \tag{7}$$

Now we present the upper bound of $\mathrm{mix}_{\mathcal{Q}}(v, u)$ for G-MHKG, also standing for the upper bound of its expressive power. Specifically, the upper bound of $\mathrm{mix}_{\mathcal{Q}}(v, u)$ is determined by $\mathbf{W}$, depth $\ell$ and $\mathbf{S} = \widehat{\mathbf{A}}^l + \widehat{\mathbf{A}}^h$, where $\widehat{\mathbf{A}}^l = \mathbf{I} - \mathbf{U}\mathbf{\Lambda}_1\mathbf{U}^\top$ and $\widehat{\mathbf{A}}^h = \mathbf{I} - \mathbf{U}\mathbf{\Lambda}_2\mathbf{U}^\top$ stand for the (weighted) adjacency matrices generated from the low pass and high pass filtering functions, respectively. We note that for the sake of simplicity, we set both $\theta_1 = \theta_2 = \mathbf{1}_N$ in Eq. (5).

**Lemma 2.** *Let $\mathcal{Q}(\mathbf{X}) = \mathbf{H}^{(\ell)}$, $\|\mathbf{W}^{(\ell-1)}\| \leq \mathrm{w}$ and $\mathbf{S} = \widehat{\mathbf{A}}^l + \widehat{\mathbf{A}}^h$. Given $u, v \in \mathcal{V}$, with $\ell$ layers, then the following holds:*

$$\left\|\frac{\partial \mathbf{h}_v^{(\ell)}}{\partial \mathbf{x}_u}\right\| \leq \mathrm{w}^\ell \, (\mathbf{S})_{v,u}^\ell \,, \tag{8}$$

*where $\| \cdot \|$ is the spectral norm of the matrix.*

We present the proof in Appendix C.2. The conclusion in Lemma 2 indicates that the incorporation of $\mathbf{W}$ serves a dual purpose: it not only plays a role in defining the model's expressive capabilities but also helps us understand the connection between the model's expressiveness and the concern of over-squashing.

**Remark 3** (HFD, over-smoothing and over-squashing). Based on the discussion in Section 4, to adapt G-MHKG to HFD dynamic, one shall require at least one of $f(\cdot)$ and $g(\cdot)$ to be *monotonically increasing* and ensure that G-MHKG always amplifies the output induced from such high pass filter determined domain (i.e., Theorem 1). Similarly, if one wants to decrease the over-squashing, it is sufficient to increase all entries of $\mathbf{S}$, so that the upper bound of mixing power increases, resulting a potentially smaller over-squashing. We note that this observation supports that any *graph re-weighting* that increases the quantity of $\mathbf{S}$ and *graph rewiring* that makes $\mathbf{S}$ denser can mitigate the over-squashing issue. Therefore it is not difficult to verify that based on the form of $\mathbf{S}$ in Lemma 2, to increase $\mathbf{S}$, it is sufficient to require $(\mathbf{U}\mathrm{diag}(\theta_1)\mathrm{e}^{f(\mathbf{\Lambda})} + \mathrm{diag}(\theta_2)\mathrm{e}^{g(\mathbf{\Lambda})}\mathbf{U}^\top)_{i,j} < \widehat{\mathbf{L}}_{i,j} \,\forall i, j$. This suggests that, similar to the model dynamics, the over-squashing issue can also be investigated through the filtering functions in spectral domain.

## 6 TRADE-OFF BETWEEN TWO ISSUES IN SPECTRAL DOMAIN

Building upon the findings from Section 4 and Section 5, this section explores a discernible trade-off between *over-smoothing* and *over-squashing*. Assuming we have two G-MHKGs namely G-MHKG(1) and G-MHKG(2), with the same assumptions in Lemma 2, additionally let two models share the same weight matrix $\mathbf{W}$. Then we are in a position to have the following conclusion.

**Lemma 3** (Trade-off). *For two G-MHKGs namely G-MHKG(1) and G-MHKG(2), if G-MHKG(1) has lower over-squashing than G-MHKG(2) i.e., $\mathrm{e}^{f_1(\mathbf{\Lambda})} + \mathrm{e}^{g_1(\mathbf{\Lambda})} < \mathrm{e}^{f_2(\mathbf{\Lambda})} + \mathrm{e}^{g_2(\mathbf{\Lambda})}$, where $f_1, g_1$ and $f_2, g_2$ are filtering functions of two models, respectively. Then on any layer i.e., from $\mathbf{H}^{(\ell-1)}$ to $\mathbf{H}^{(\ell)}$. Taking $\mathbf{H}^{(\ell-1)}$ as the initial input feature at layer $\ell$, we have the following inequality in terms of Dirichlet energy of two models: $\mathbf{E}(\mathbf{H}^{(\ell)})_1 < \mathbf{E}(\mathbf{H}^{(\ell)})_2$. In words, more feature smoothing effect is induced from lower over-squashing model G-MHKG(1).*

We include the proof in Appendix C.3. Our conclusion can be applied to single-scale GNNs. To gain a clear understanding of the trade-off described in Lemma 3, one can consider that since the sum of the filtering functions of G-MHKG(2) over G-MHKG(1) indicating a higher Dirichlet energy of the node features, and thus G-MHKG(2) produces less smoothing than G-MHKG(1). Meanwhile, the higher values of eigenvalues from G-MHKG(2) also increase model's over-squashing based on Lemma 2 and Remark 3. These observations directly suggest that imposing more sharpening effect in a model to prevent over-smoothing will lead to more over-squashing.

## 7 TIME MANIPULATION: HOW TO HANDLE TWO ISSUES AND BEYOND

While we have illustrated the fundamental relationship (trade-off) between over-smoothing and over-squashing, whether GNN models can handle both issues naturally becomes the next question. Specifically, one shall require a GNN to be HFD to avoid over-smoothing and increasing the quantities in $\mathbf{S}$ to decrease over-squashing. In terms of G-MHKG, one can see that due to the property of exponential function, it is not possible to have the summation of the diagonal position of $\mathrm{diag}(\theta_1)\mathrm{e}^{f(\mathbf{\Lambda})} + \mathrm{diag}(\theta_2)\mathrm{e}^{g(\mathbf{\Lambda})} = 0$ unless we set $\theta_1 = \theta_2 = 0$ under the situation that $\mathrm{e}^{f(\mathbf{\Lambda})} \neq \mathrm{e}^{g(\mathbf{\Lambda})}$, or $\theta_1 = -\theta_2$ when $\mathrm{e}^{f(\mathbf{\Lambda})} = \mathrm{e}^{g(\mathbf{\Lambda})}$. Nonetheless, to satisfy the condition, we delay the occurrence of HFD by setting $\theta_1 = \theta_2 = 0$ when $\lambda = 0$, and our result is summarized in the following theorem.

**Theorem 2** (Delayed HFD (D-HFD)). *G-MHKG is capable of handling both two issues with HFD dynamic and non-increasing over-squashing. Specifically, let $k$ be the number of connected compo-*

*nents of $\mathcal{G}$ and $\theta_1$, $\theta_2 \geq 0$ then both two issues can be sufficiently handled by setting* $\mathrm{diag}(\theta_1)_{i,i} = \mathrm{diag}(\theta_2)_{i,i} = 0$ $i \in [1, k]$ *and* $\mathrm{diag}(\theta_1)\mathrm{e}^{f(\mathbf{\Lambda}_{i,i})} + \mathrm{diag}(\theta_2)\mathrm{e}^{g(\mathbf{\Lambda}_{i,i})} < \mathbf{\Lambda}_{i,i}$ $i \in [k+1, N]$.

We included the detailed proof in Appendix C.4 with additional discussions and clarifications. Importantly from Theorem 2, the reason why we consider zero eigenvalues of the graph is to sufficiently decrease the over-squashing, one shall require the filtered eigenvalues are not larger than the graph spectra. Therefore when eigenvalues are 0, the sufficient condition is to require the filtering results equal to 0 to maintain the over-squashing level. Figure 2 shows the comparison between different kinds of filtering functions in regard to the effect on over-smoothing and over-squashing. In addition, although Theorem 2 show how a D-HFD model handle two issues, this conclusion is more applicable to heterophily graphs according to Remark 1. However, as we will show in the next theorem, it is not possible for the model to be LFD and decrease over-squashing.

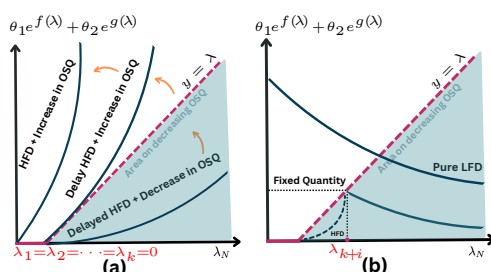

Figure 2: The figure on the left represents different types of HFD filtering outcomes and the trade-off between two issues. One can check that to induce more sharpening (filtering function from bottom to top), the model will suffer more from OSQ. The figure on the right illustrates the situation described in Theorem 3.

**Theorem 3.** *Suppose $\theta_1$ and $\theta_2 \geq 0$, then it is impossible for the model to be LFD and decrease OSQ. In other words, there must exist at least one $\mathbf{\Lambda}^*$ such that* $\mathrm{diag}(\theta_1)\mathrm{e}^{f(\mathbf{\Lambda}^*)} + \mathrm{diag}(\theta_2)\mathrm{e}^{g(\mathbf{\Lambda}^*)}$ *is (1) monotonically increasing (HFD) or (2) with constant quantity (thus not LFD) or (3) monotonically decreasing with image greater than $\mathbf{\Lambda}$ thus OSQ is increased.*

The proof of theorem is in Appendix C.5. Figure 2 (b) illustrates the situation in Theorem 3. The model with the dynamic shown in Figure 2 (b) (i.e., first $k$ components being 0 + dashed HFD + LFD) is not a LFD and in fact asymptotically dominated by the HFD part, thereby according to Definition 1 and Lemma 1, the model is not L/HFD.

**Relation to existing works**  Compared to the existing work (Giraldo et al., 2022) that claimed both issues can not be alleviated simultaneously, our conclusion shows that once the model is HFD, it is possible to handle both issues. This suggests the effectiveness for incorporating multi-scale GNNs in terms of investigating the over-smoothing issue via model dynamics. Furthermore, although similar motivation for inducing sharpening effect to the node features was explored (Choi et al., 2023), its effect was only verified empirically. Our analysis under the scope of dominant dynamic and maximal mixing of the function learned from GNNs pave the path of evaluating well-known GNN issues in a unified platform. Moreover, recent studies (Nguyen et al., 2023) also attempted to mitigate both issues via graph surgery by dropping highly positive cured edges which often lead to over-smoothing issue and rewiring the communities connected with very negatively curved edges which is responsible for the over-squashing issue (Topping et al., 2021). Although the relationship between the graph edge curvature and spectra was explored (Bauer et al., 2011), a detailed comparison between spatial (curvature based surgery) and spectral method is still wanted, especially on how the spatial surgery affects the eigen-distribution of the graph spectra (Shi et al., 2023a).

## 8 EXPERIMENT

In this section, we show a variety of numerical tests for MHKG and G-MHKG. Specifically, Section 8.1 verifies how a controlled model dynamic enhances model's adaption power on homophily and heterophily graphs. In Section 8.2 we will compare model performance under four different dynamics to illustrate how D-HFD dynamic assists model in handling two issues. Furthermore, Section 8.3 will show the performance (node classification) of MHKG and G-MHKG via real-world citation networks as well as large-scale graph datasets. We include more discussions (i.e., computational complexity) and ablation studies in Appendix D. All experiments were conducted using PyTorch on Tesla V100 GPU with 5,120 CUDA cores and 16GB HBM2 mounted on an HPC cluster. The source code can be found in `https://anonymous.4open.science/r/G_MHKG_accept`.

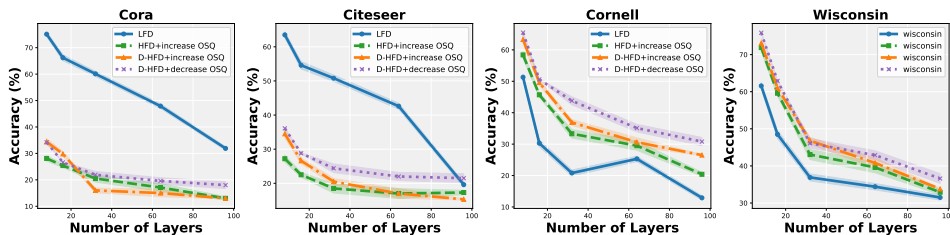

Figure 4: Results on model with different dynamics. Number of layers from 8, 16, 32, 64, and 96.

**Experiment Setup** For the settings in MHKG, we followed form of MHKG defined in Eq. (3) by setting $f(\mathbf{\Lambda}) = \mathbf{U}(e^{\mathbf{\Lambda}})\mathbf{U}^{\top}$. We note that in general the form of $f$ can be any *monotonic positive* function on $\mathbf{\Lambda}$, here we only choose the one in Eq. (3) for the consistency reason. In regarding to G-MHKG, we assign one initial warm-up coefficient $\gamma > 1$ that is multiplied with $f(\widehat{\mathbf{L}})$ if the graph is homophily otherwise on $g(\widehat{\mathbf{L}})$ if a graph is heterophily. We note that this operation aims to ensure G-MHKG to induce more smoothing/sharpening effect for fitting different types of graphs according to Remark 1. We then re-scale the result of the filtering functions back to $[0, 2]$ so that the assumption of Lemma 2 holds. We included **Cora, Citeseer, Pubmed** for homophily datasets and applied public split (Pei et al., 2020) and **Wisconsin, Texas, Cornell** as heterophily datasets with 60% for training, 20% for testing and validation. In addition, we also included one large-scale graph dataset **ogbn-arxiv** to illustrate model scalability for large scale datasets. The summary statistics of all included benchmarks as well as the model hyper-parameter search space are included in Appendix D.1. We set the maximum number of epochs of 200 for citation networks and 500 for **ogbn-arxiv**. The average test accuracy and its standard deviation come from 10 runs.

## 8.1 CONTROLLED MODEL DYNAMIC

In this section, we verify Theorem 1 by assigning different quantities of $\zeta$ so that G-MHKG can be L/HFD. Specifically, we fixed $\theta_1 = \mathbf{1}_N, \mathbf{2}_N$, and $\mathbf{3}_N$ and set the value of $\zeta$ from 0.5 to 3 with the unit of change as 0.5 so that model dynamics changed from HFD to LFD (i.e., $\zeta = 0.5$). All other hyperparameters were fixed across the models. We conducted the experiment on **Cora** and **Texas**, and Figure 3 shows the changes in the learning accuracy on both datasets. It is clear to see that with the increase of $\zeta = \theta_1/\theta_2$, model's dynamics change from LFD to HFD, resulting in more adaption power from homophily to heterophily graphs.

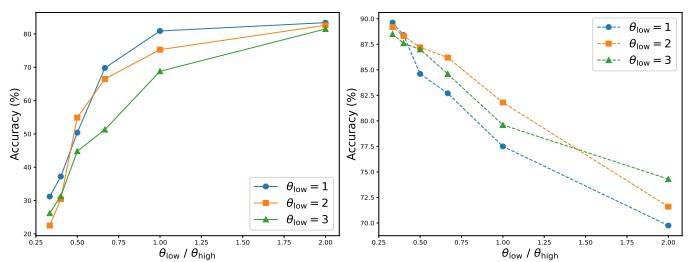

Figure 3: Model Accuracy(%) via different ratios between filtering matrices. Left: Accuracy on **Cora**, Right: Accuracy on **Texas**.

## 8.2 HANDLING BOTH ISSUES WITH D-HFD AND ABLATION

In this section, we show how G-MHKG with D-HFD dynamic handles both over-smoothing and over-squashing issues. Specifically, we compare G-MHKG's performance via the following four dynamics: LFD ($\theta_1 > \theta_2 > 0$), HFD + increase in OSQ (sufficient large of $\theta_2$), D-HFD + increase in OSQ ($\theta_2$ sufficient large and first $k$ components of both $\theta_1$ and $\theta_2$ are 0) and D-HFD + decrease in OSQ (Theorem 2). We include more details on the model setup for inducing the aforementioned dynamics in Appendix D.2. We select two homophily graphs (**Cora** with $k = 25$ and **Citeseer** $k = 115$) and heterophily graphs (**Cornell** $k = 6$ and **Wisconsin** $k = 16$). We tested G-MHKG with $8, 16, 32, 64$ and 96 layers to illustrate the asymptotic behavior of the model via different dynamics. We note that this experiment can also be served as an ablation study which proves the advantage of D-HFD for heterophily graphs. Figure 4 illustrates the accuracy comparison between four dynamics. One can see from Figure 4 that for homophily graphs, LFD (blue) models always present the top

performances compared to other three dynamics, followed by the dynamic of D-HFD + decreasing OSQ (violet) as such dynamic assists model to handle over-squashing issue. The last two ranks are achieved by D-HFD + increase OSQ (orange) and HFD + increase OSQ (green) with the former dynamic not increase OSQ with the first $k$ eigenvalues and latter suffering from both issues. For heterophily graphs, with the best accuracy achieved via D-HFD + decreased OSQ across all layers, and the worst outcome in LFD which is unnecessary for homophily graphs.

## 8.3 RESULTS OF NODE CLASSIFICATION ON REAL-WORLD DATA

We include the introduction of the baseline models and the reason of choosing them in Appendix D.1. We design MHKG and G-MHKG with two convolution layers followed by softmax activation function. The results of node classification are included in Table 1, and all baseline results are listed according to the existing publications. One can check that G-MHKG show remarkable performance on both homophily and heterophily graphs.

Table 1: Performance on node classification using public split. Top two in **bold**.

| Methods | Cora | Citeseer | Pubmed | Cornell | Texas | Wisconsin | Arxiv |
|---------|------|----------|--------|---------|-------|-----------|-------|
| MLP | 55.1 | 59.1 | 71.4 | **91.3±0.7** | **92.3±0.7** | **91.8±3.1** | 55.0±0.3 |
| GCN | 81.5±0.5 | 70.9±0.5 | 79.0±0.3 | 66.5±13.8 | 75.7±1.0 | 66.7±1.4 | **72.7±0.3** |
| GAT | 83.0±0.7 | **72.0±0.7** | 78.5±0.3 | 76.0±1.0 | 78.8±0.9 | 71.0±4.6 | 72.0±0.5 |
| GIN | 78.6±1.2 | 71.4±1.1 | 76.9±0.6 | 78.0±1.9 | 74.6±0.8 | 72.9±2.5 | 64.5±2.5 |
| HKGCN | 81.9±0.9 | 72.4±0.4 | 79.9±0.3 | 74.2±2.1 | 82.4±0.7 | 85.5±2.7 | 69.6±1.7 |
| GRAND | 82.9±1.4 | 70.8±1.1 | 79.2±1.5 | 72.2±3.1 | 80.2±1.5 | 86.4±2.7 | 71.2±0.2 |
| UFG | **83.3±0.5** | 71.0±0.6 | 79.4±0.4 | 83.2±0.3 | 82.3±0.9 | **91.9±2.1** | **72.6±0.1** |
| SJLR | 81.3±0.5 | 70.6±0.4 | 78.0±0.3 | 71.9±1.9 | 80.1±0.9 | 66.9±2.1 | 72.0±0.4 |
| MHKG | 82.8±0.2 | 71.6±0.1 | 78.9±0.3 | 86.2±0.6 | 84.5±0.3 | 88.9±0.3 | 72.1±0.6 |
| G-MHKG | **83.5±0.2** | **72.8±0.2** | **80.1±0.4** | **90.2±0.9** | **89.6±0.6** | 91.2±1.5 | 72.4±0.3 |

## 8.4 DISCUSSION ON LIMITATION AND WAYS FORWARD

**Limitation on L/HFD, why there is an accuracy drop?** We found that there is a considerable accuracy drop between the results in Table. 1 and model with four different dynamics and different layers. This observation suggests that although L/HFD are proved to be more suitable for homo/heterophily graphs, under large number of layers, i.e., GNNs propagates feature information from large hops, the measure of homophily level (Zhu et al., 2021) become powerless, since such level varies through hops and thus requiring GNNs to induce a layer-wise rather than overall dynamics when the layer number is high. This observation aligns with the motivation of the recent work in (Lee et al., 2023).

**The measure of OSQ** Lemma 2 establishes an upper bound for the node sensitivity measure on the issue of over-squashing. While Lemma 2 offers a necessary condition, it does not guarantee that G-MHKG possesses the desired mixing power. Hence, a complementary lower bound that provides a sufficient condition becomes essential. Thereby in scenarios where both over-smoothing and over-squashing need to be considered, an adjusted conclusion is sought after.

## 9 CONCLUSION

In this paper, we explored the underlying relationship between three fundamental issues of GNNs: over-smoothing, over-squashing, and expressive power via proposed G-MHKG induced by reversing the time direction of so-called graph heat equation. We revealed the roles of the filtering and weight matrices via gradient flow and maximal mixing perspectives to illustrate their capability of controlling aforementioned issues. Furthermore, we show that under mild conditions on the filtering equations in G-MHKG, there is a fundamental trade-off between over-smoothing and over-squashing via graph spectral domain. We further showed that our proposed model is capable of handling both issues and own its advantage in terms of mixing smoothing and sharpening effects compared to single-scale GNNs. While we have shown superior performance of G-MHKG empirically, many unknown issues we have listed still inspire us to explore further. In future works, we will attempt to discover the necessary conditions for one GNN that is capable of handling all mentioned issues.

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

# A  RELATED WORKS

## A.1  HEAT OPERATOR AND HEAT KERNEL

In this section, we show a more detailed introduction of heat operator and heat kernel. Let $\mathcal{M}$ be a compact Riemannian manifold possibly with boundary. The so-called heat diffusion process on the manifold is denoted by the heat equation: $\Delta_{\mathcal{M}} u(x, t) = -\frac{\partial u(x,t)}{\partial t}$, where $\Delta_{\mathcal{M}}$ is known as Laplace-Beltrami operator of $\mathcal{M}$. Given the initial heat distribution $q : \mathcal{M} \to \mathbb{R}$, one can let $\mathcal{S}^{(t)}(q)$ satisfies the heat equation for all $t$ and $\mathrm{Lim}_{t \to 0} \mathcal{S}^{(t)}(q) = q$. Then $\mathcal{S}^{(t)}$ is called the heat operator. One can check that both $\Delta_{\mathcal{M}}$ and $\mathcal{S}^{(t)}$ maps a real function defined on $\mathcal{M}$ to another function, and the transaction between them can be denoted as $\mathcal{S}^{(t)} = \mathrm{e}^{-t\Delta_{\mathcal{M}}}$. Therefore, both two operators share the same eigen-functions, and if $\lambda_i$ is the $i$-th eigenvalue of $\Delta_{\mathcal{M}}$ then $\mathrm{e}^{-t\lambda_i}$ is the $i$-th eigenvalue of $\mathcal{S}^{(t)}$. Based on the work in (Hsu, 2002; Sun et al., 2009), for any $\mathcal{M}$, there exists a function $k^{(t)}(x, y) : \mathbb{R}^+ \times \mathcal{M} \times \mathcal{M} \to \mathbb{R}$ such that:

$$\mathcal{S}^{(t)}(q(x)) = \int_{\mathcal{M}} k^{(t)}(x, y) q(y) dy, \tag{9}$$

where $dy$ is the volume form at $y \in \mathcal{M}$. The minimum function $k^{(t)}(x, y)$ that satisfies Eq. (9) is called the heat kernel which can be interpreted as the amount of heat that transacts from the point $x$ to $y$ during the time period $t$. For any given continuous time period, the heat kernel shows a continuous smoothing diffusion process of the temperature on the manifold. Now we analogize this notion to the discrete graph domain where we replace the $\Delta_{\mathcal{M}}$ to the graph (normalized) Laplacian $\widehat{\mathbf{L}}$, and for any $t > 0$ the heat kernel of $\mathcal{G}$ can be served as the fundamental solution of the graph heat equation:

$$\frac{\partial \mathbf{H}^{(t)}}{\partial t} = -\widehat{\mathbf{L}} \mathbf{H}^{(t)}, \tag{10}$$

in which we denote the $\mathbf{H}^{(t)}$ as the feature representation at a specific iteration time $t$ of GNN with $\mathbf{H}^{(0)} = \mathbf{X}$. The solution of the graph heat equation, denoted as $\mathbf{H}^{(t)}$, is given by $\mathbf{H}^{(t)} = \mathrm{e}^{-t\widehat{\mathbf{L}}} \mathbf{H}^{(0)}$. It is well-known that the application of Euler discretization leads to the propagation of the linear GCN models (Kipf & Welling, 2016; Wu et al., 2019) and this process is with the name of Laplacian smoothing (Chung, 1997). The characteristic of the solution is indeed governed by the so-called Heat Kernel, denoted by $\widehat{\mathbf{H}}_t = \mathrm{e}^{-t\widehat{\mathbf{L}}}$. The heat kernel defines a continuous-time random walk, and defines a semi-group, i.e., $\mathbf{H}^{(t+s)} = \mathbf{H}^{(t)} \cdot \mathbf{H}^{(s)}$ for any $t, s \geq 0$, and $\lim_{t \to \infty} \mathbf{H}^{(t)} = \mathbf{I}$ (Chung, 1997). This fact indicates non-distinguishment for the nodes with same degrees, known as over-smoothing.

## A.2  MULTI-SCALE GNNS AND GRAPH FRAMELETS

The notion of so-called multi-scale GNNs can be traced back to the application of graph wavelet analysis (Crovella & Kolaczyk, 2003) for traffic forecasting. The work Maggioni & Mhaskar (2008) applied the polynomials of a differential operator to build multi-scale transforms. The spectral graph wavelet transforms (Hammond et al., 2011) define the graph spectrum from a graph's Laplacian matrix, where the scaling function is approximated by the Chebyshev polynomials. Dong (2017) approximated piece-wise smooth functions with undecimated tight wavelet frames. Fast decomposition and reconstruction become possible with the filtered Chebyshev polynomial approximation and proper design of filter banks. Meanwhile, as an extension of the wavelet analysis, fast tight framelet filter bank transforms on quadrature-based framelets are explored on graph domain (Wang & Zhuang, 2019; Zheng et al., 2022). Based on these constructive works, multi-scale GNNs have shown superior performances in terms of both node and graph level classification tasks. Specifically, Zheng et al. (2021) formally deployed fast and tight framelet decomposition and reconstruction on graph, resulting as a general form of multi-scale graph framelet convolution. Motivated by the idea of adapting framelet onto direct graphs, Zou et al. (2022) proposed SVD based graph framelets, followed by Yang et al. (2022) who further introduced quasi-framelet so that more flexible filtering functions can be utilized. In fact, one can consider the learning process of multi-scale GNNs (including graph framelet) as learning multiple kernels that conditional to the feature aggregation. Thereby including more graph related information (i.e., curvature (Shi et al., 2023a)) to the propagation often leads to a better learning outcomes.

### A.3 CURRENT STUDY IN UNDERSTANDING ON BOTH ISSUES

Unlike the over-smoothing issue, which has been established for years (Li et al., 2018), the over-squashing issue, however, was only identified and analyzed recently (Topping et al., 2021; Alon & Yahav, 2020). Specifically, the majority of the GNN researches quantified the phenomenon of over-smoothing through the measures of variation of the node features (i.e., Dirichlet energy or its variants) generated from the propagation of GNNs. The connection between over-squashing and so-called negative Forman curvature has been established recently (Topping et al., 2021). Accordingly, these two different paths of investigating two issues lead GNNs towards different form of propagations, causing lack of unified manner of exploring both problems. The recent research that consider both problems can be found in (Giraldo et al., 2022) in which a curvature based rewiring method is deployed to mitigate both issues, and claimed that both issue can not be solved simultaneously, which paved the path of investigating both issues in an unified platform. Similar research using the sign of curvature to link both issues can also be found in (Nguyen et al., 2023) in which a novel curvature flow (Batch Ollivier-Ricci Flow) method is proposed to mitigate both issues.

## B LIST OF GNNs AS SPECIAL FORM OF THE PROPOSED MODEL

In this section, we list the relationship between popular GNNs and G-MHKG. Recall that the propagation of G-MHKG is:

$$\mathbf{H}^{(\ell)} = \mathbf{U}\mathrm{diag}(\theta_1)\mathrm{e}^{f(\mathbf{\Lambda})}\mathbf{U}^\top\mathbf{H}^{(\ell-1)}\mathbf{W}^{(\ell-1)} + \mathbf{U}\mathrm{diag}(\theta_2)\mathrm{e}^{g(\mathbf{\Lambda})}\mathbf{U}^\top\mathbf{H}^{(\ell-1)}\mathbf{W}^{(\ell-1)}. \quad (11)$$

It is not hard to verify that, by setting either $\theta_1$ or $\theta_2 = 0$, we recover the GRAND (Chamberlain et al., 2021a) with one additional channel mixing matrix $\mathbf{W}$. Letting $g(\mathbf{\Lambda}) = 0$, $\theta_2 = \mathbf{c}_N$, where $\mathbf{c}_N$ is a $N$-dimensional vector with all $c$ as an arbitrary constant, we have the model equivalent to GRAND++ (Thorpe et al., 2021) with a layer dependent source term that is $\widetilde{\mathbf{H}} = \mathbf{U}\mathrm{diag}(\theta_2)\mathbf{U}^\top\mathbf{H}\mathbf{W}$. Furthermore, if one represents G-MHKG via spatial message passing followed by Lemma 2 i.e., $\mathbf{H}^{(\ell)} = \mathbf{S}\mathbf{H}^{(\ell-1)}\mathbf{W}^{(\ell-1)}$, where $\mathbf{S} = \widehat{\mathbf{A}}^l + \widehat{\mathbf{A}}^h$ and $\widehat{\mathbf{A}}$ are the corresponding adjacency matrix. Then one can obtain the reweighting matrix $\boldsymbol{\xi} = \mathbf{S} \oslash \widehat{\mathbf{A}}$ and in this case, G-MHKG is same as those GNNs with adjacency reweighting schemes such as GAT (Veličković et al., 2018) in single-scale and Curvature framelet GCN (Shi et al., 2023a) as multi-scale.

More precisely, let $g(\mathbf{\Lambda}) = 0$, the propagation of G-MHKG becomes

$$\mathbf{H}^{(\ell)} = \mathbf{U}\mathrm{diag}(\theta_1)\mathrm{e}^{f(\mathbf{\Lambda})}\mathbf{U}^\top\mathbf{H}^{(\ell-1)}\mathbf{W}^{(\ell-1)} + \mathbf{U}\mathrm{diag}(\theta_2)\mathbf{U}^\top\mathbf{H}^{(\ell-1)}\mathbf{W}^{(\ell-1)}$$

$$= \mathbf{U}\mathrm{diag}(\theta_1)\mathrm{e}^{f(\mathbf{\Lambda})}\mathbf{U}^\top\mathbf{H}^{(\ell-1)}\mathbf{W}^{(\ell-1)} + \widetilde{\mathbf{H}}^{(\ell-1)}, \quad (12)$$

in which the first term can be treated as the special graph neural diffusion model and the last term is a layer dependent node feature, indicating a combination of the diffusion and a source term, yielding a special form of GRAND++. Additionally, let $\mathbf{S} = \widehat{\mathbf{A}}^l + \widehat{\mathbf{A}}^h$, where $\widehat{\mathbf{A}}^l = \mathbf{I} - \mathbf{U}\mathbf{\Lambda}_1\mathbf{U}^\top$ and $\widehat{\mathbf{A}}^h = \mathbf{I} - \mathbf{U}\mathbf{\Lambda}_2\mathbf{U}^\top$ stand for the (weighted) adjacency matrices generated from the low pass and high pass filtering functions (i.e., $f(\cdot)$, $g(\mathrm{cot})$), respectively with $\mathbf{\Lambda}_1 = f(\mathbf{\Lambda}) = \mathrm{diag}(\{\mathrm{e}^{f(\lambda_i)}\}_{i=1}^N)$ and $\mathbf{\Lambda}_2 = g(\mathbf{\Lambda}) = \mathrm{diag}(\{\mathrm{e}^{g(\lambda_i)}\}_{i=1}^N)$. Then one can show that the adjacency information ($\mathbf{S} = \widehat{\mathbf{A}}^l + \widehat{\mathbf{A}}^h$) that propagated in G-MHKG is indeed the sum of two weighted adjacency matrices while preserving the graph connectivity. Therefore, with $\boldsymbol{\xi} = \mathbf{S} \oslash \widehat{\mathbf{A}}$, one can show that G-MHKG is in fact align with GNNs enhanced with reweighting scheme (i.e., GAT and curvature framelet GCN).

## C FORMAL PROOFS

### C.1 PROOF OF THEOREM 1.

In this section, we show the proof of Theorem 1 which indicates that G-MHKG can induce both energy dynamics under different settings of the filtering matrices $\mathrm{diag}(\theta_1)$ and $\mathrm{diag}(\theta_2)$.

**Theorem 4** (Repeat of Theorem 1). *G-MHKG can induce both LFD and HFD dynamics. Specifically, let $\theta_1 = \mathbf{1}_N$ and $\theta_2 = \zeta\mathbf{1}_N$ with a positive constant $\zeta$. Then, with sufficient large $\zeta$ ($\zeta > 1$) so*

*that* $\mathrm{e}^{f(\mathbf{\Lambda})} + \zeta \mathrm{e}^{g(\mathbf{\Lambda})}$ *is monotonically increasing on spectral domain, G-MHKG is HFD. Similarly, if* $0 < \zeta < 1$ *and is sufficient small such that* $\mathrm{e}^{f(\hat{\mathbf{\Lambda}})} + \zeta \mathrm{e}^{g(\mathbf{\Lambda})}$ *is monotonically decreasing, the model is LFD.*

*Proof.* As we aim to prove the existence of the filtering matrices for L/HFD. Without loss of generality, we show the example with Theorem 1 holds. Recall that G-MHKG has the propagation rule as:

$$\mathbf{H}^{(\ell)} = \mathbf{U}\mathrm{diag}(\theta_1)\mathrm{e}^{f(\mathbf{\Lambda})}\mathbf{U}^{\top}\mathbf{H}^{(\ell-1)}\mathbf{W}^{(\ell-1)} + \mathbf{U}\mathrm{diag}(\theta_2)\mathrm{e}^{g(\mathbf{\Lambda})}\mathbf{U}^{\top}\mathbf{H}^{(\ell-1)}\mathbf{W}^{(\ell-1)}$$
$$= \mathcal{W}_l\mathbf{H}^{(\ell-1)}\mathbf{W}^{(\ell-1)} + \mathcal{W}_h\mathbf{H}^{(\ell-1)}\mathbf{W}^{(\ell-1)} \tag{13}$$

where we let $\mathcal{W}_l = \mathbf{U}\mathrm{diag}(\theta_1)\mathrm{e}^{f(\mathbf{\Lambda})}\mathbf{U}^{\top}$ and $\mathcal{W}_h = \mathbf{U}\mathrm{diag}(\theta_2)\mathrm{e}^{g(\mathbf{\Lambda})}\mathbf{U}^{\top}$, respectively. Plugging in the setting of $\theta_1$ and $\theta_2$, we have $\mathcal{W}_l = \mathbf{U}\mathrm{e}^{f(\mathbf{\Lambda})}\mathbf{U}^{\top}$ and $\mathcal{W}_h = \mathbf{U}\zeta\mathrm{e}^{g(\mathbf{\Lambda})}\mathbf{U}^{\top}$, respectively. Furthermore, as in G-MHKG, every step we applied the same operation on the feature from the last step, one can let $t = m\tau$ and $\tau = 1$ for $m$ time propagation and each propagation takes the step as 1. Together with the conditions on $\theta_1$ and $\theta_2$, Eq. (13) can be further expressed as:

$$\mathrm{vec}(\mathbf{H}^{(m\tau)}) = \tau^m \left(\mathbf{W} \otimes (\mathcal{W}_\ell + \zeta\mathcal{W}_h)\right)^m \mathrm{vec}(\mathbf{H}(0))$$
$$= \tau^m \sum_{i,k} \left(\lambda_k^W \left(\mathrm{e}^{f(\lambda_i)} + \zeta\mathrm{e}^{g(\lambda_i)}\right)\right)^m \mathrm{c}_{k,i}(0)\phi_k^W \otimes \mathbf{u}_i, \tag{14}$$

where we let $\{(\lambda_k^W, \phi_k^W)\}_{k=1}^c$ as the eigen-pairs of $\mathbf{W}$ and denote $\mathrm{c}_{k,i}(0) = \langle\mathrm{vec}(\mathbf{H}(0)), \phi_k^W \otimes \mathbf{u}_i\rangle$. It is worth noting that, without loss of generality, we set $\mathbf{W}^{(\ell-1)} = \mathbf{W}^{(\ell-2)} = \cdots = \mathbf{W}$, suggesting a fixed weight matrix for all layers.

We now show that according to different quantity of $\zeta$, the dynamic in Eq. (14) can be L/HFD. Then with sufficient large of $\zeta$ so that $\mathrm{e}^{f(\hat{\mathbf{L}})} + \zeta\mathrm{e}^{g(\hat{\mathbf{L}})}$ is a *monotonically increase*, one can find that the *unique* maximal image of $|\lambda_k^W \left(\mathrm{e}^{f(\lambda_i)} + \zeta\mathrm{e}^{g(\lambda_i)}\right)|$ at frequency $\rho_{\hat{\mathbf{L}}}$. Furthermore, denote $\delta_{\mathrm{HFD}} := \lambda_k^W \left(\mathrm{e}^{f(\rho_{\hat{\mathbf{L}}})} + \zeta\mathrm{e}^{g(\rho_{\hat{\mathbf{L}}})}\right)$ and $\zeta > 1$ sufficient large such that for all $i$ where $\lambda_i \neq \rho_{\hat{\mathbf{L}}}$, $|\lambda_k^W \left(\mathrm{e}^{f(\lambda_i)} + \zeta\mathrm{e}^{g(\lambda_i)}\right)| < \delta_{\mathrm{HFD}}$ holds. Then the dynamic is HFD and the dominant frequency is $\rho_{\hat{\mathbf{L}}}$. One can also verify the model can be LFD by utilizing the same approach.

More precisely, based on the Definition 1 and Lemma 1, let $\delta := \max_{i:\lambda_i \neq \rho_L} |\lambda_k^W \left(\mathrm{e}^{f(\lambda_i)} + \zeta\mathrm{e}^{g(\lambda_i)}\right)|$, also denote $\mathbf{P}_\rho = \sum_k (\phi_k^W \otimes \mathbf{u}_\rho)(\phi_k^W \otimes \mathbf{u}_\rho)^{\top}$ where $\mathbf{u}_\rho$ is the eigenvector of $\hat{\mathbf{L}}$ associated with eigenvalue $\rho_{\hat{\mathbf{L}}}$ (assuming the eigenvalue $\rho_{\hat{\mathbf{L}}}$ is simple). Then we can decompose Eq. (14) as

$$\mathrm{vec}\big(\mathbf{H}(m\tau)\big)$$
$$= \tau^m \sum_k \delta_{\mathrm{HFD}}^m \mathrm{c}_{k,\rho_L}(0)\phi_k \otimes \mathbf{u}_\rho + \tau^m \sum_k \sum_{i:\lambda_i \neq \rho_L} \left(\left(\lambda_k^W(\mathrm{e}^{f(\lambda_i)} + \zeta(\mathrm{e}^{g(\lambda_i)}))\right)\right)^m \mathrm{c}_{k,i}(0)\phi_k \otimes \mathbf{u}_i$$
$$\leq \tau^m \delta_{\mathrm{HFD}}^m (\mathbf{P}_\rho\mathrm{vec}\big(\mathbf{H}^{(0)}\big) + \sum_k \sum_{i:\lambda_i \neq \rho_L} \left(\frac{\delta}{\delta_{\mathrm{HFD}}}\right)^m \mathrm{c}_{k,i}(0)\phi_k \otimes \mathbf{u}_i,$$

where $\delta < \delta_{\mathrm{HFD}}$. By normalizing the results, we obtain $\frac{\mathrm{vec}\big(\mathbf{H}(m\tau)\big)}{\|\mathrm{vec}\big(\mathbf{H}(m\tau)\big)\|} \rightarrow \frac{\mathbf{P}_\rho(\mathrm{vec}(\mathbf{H}(0)))}{\|\mathbf{P}_\rho\mathrm{vec}(\mathbf{H}(0))\|}$, as $m \rightarrow \infty$, where the latter is a unit vector $\mathbf{h}_\infty$ satisfying $(\mathbf{I}_c \otimes \hat{\mathbf{L}})\mathbf{h}_\infty = \rho_{\hat{\mathbf{L}}}\mathbf{h}_\infty$. This suggests the dynamic is HFD according to Definition 1 and Lemma 1. $\qquad\square$

**Remark 4.** The key of proving Theorem 1 is to evaluate the relative importance of the two terms in the sum filtering functions $\left(\mathrm{e}^{f(\mathbf{\Lambda})} + \zeta\mathrm{e}^{g(\mathbf{\Lambda})}\right)$. It is worth noting that if one goes specifically to the monotonicity of $f(\cdot)$ and $g(\cdot)$, i.e. both $f(\cdot)$ and $g(\cdot)$ *monotonically increasing*, then regardless of the choice of $\zeta$, the model is always HFD. When $f(\cdot)$ is monotonically decreasing and $g(\cdot)$ is monotonically increasing, with sufficient large of $\zeta$, the model remains HFD. Last, if both $f(\cdot)$ and $g(\cdot)$ are monotonically decreasing, then regardless of the quantity of $\zeta$ the model is always LFD.

## C.2 PROOF OF LEMMA 2

In this section, we prove the upper bound of the maximal mixing of G-MHKG included in Lemma 2:

**Lemma 4** (Repeat of Lemma 2). *Let $\mathcal{Q}(\mathbf{X}) = \mathbf{H}^{(\ell)}$, $\|\mathbf{W}^{(\ell-1)}\| \leq \mathrm{w}$ and $\mathbf{S} = \widehat{\mathbf{A}}^l + \widehat{\mathbf{A}}^h$. Given $u, v \in \mathcal{V}$, with $\ell$ layers, then the following holds:*

$$\left\| \frac{\partial \mathbf{h}_v^{(\ell)}}{\partial \mathbf{x}_u} \right\| \leq \mathrm{w}^\ell \left( \mathbf{S} \right)_{v,u}^\ell, \tag{15}$$

*where $\| \cdot \|$ is the spectral norm of the matrix.*

*Proof.* We note that the proof for the conclusion in the case of single-scale GNN (i.e. $\mathbf{S} = \widehat{\mathbf{A}}$) is done in (Shi et al., 2022). Here we directly generalize the proof to multi-scale case for self-completeness. First recall that G-MHKG is with the propagation rule as:

$$\begin{aligned} \mathbf{H}^{(\ell)} &= \mathbf{U}\mathrm{diag}(\theta_1)\mathrm{e}^{f(\mathbf{\Lambda})}\mathbf{U}^\top \mathbf{H}^{(\ell-1)}\mathbf{W}^{(\ell-1)} + \mathbf{U}\mathrm{diag}(\theta_2)\mathrm{e}^{g(\mathbf{\Lambda})}\mathbf{U}^\top \mathbf{H}^{(\ell-1)}\mathbf{W}^{(\ell-1)} \\ &= \mathbf{U}\left( \mathrm{diag}(\theta_1)\mathrm{e}^{f(\mathbf{\Lambda})} + \mathrm{diag}(\theta_2)\mathrm{e}^{g(\mathbf{\Lambda})} \right) \mathbf{U}^\top \mathbf{H}^{(\ell-1)}\mathbf{W}^{(\ell-1)} \\ &= \widehat{\mathbf{L}}^* \mathbf{H}^{(m-1)}\mathbf{W}^{(m-1)}, \end{aligned} \tag{16}$$

where $\mathrm{diag}(\theta_1)\widehat{\mathbf{L}}^l$ and $\mathrm{diag}(\theta_2)\widehat{\mathbf{L}}^h$ stand for the spectral filtering process on the corresponding Laplacian induced from the filtering functions. Based on the relationship between spatial and spectral GNNs, one can further denote the propagation in Eq. (16) with a spatial based form as: $\mathbf{H}^{(\ell)} = \mathbf{S}\mathbf{H}^{(\ell-1)}\mathbf{W}^{(\ell-1)}$, where $\mathbf{S} = \widehat{\mathbf{A}}^l + \widehat{\mathbf{A}}^h$. Now we see that $\mathbf{h}_v^{(\ell)} = (\mathbf{W}^{(\ell-1)})^\top (\mathbf{H}^{(\ell-1)})^\top \mathbf{s}_v = \sum_{i=1}^N s_{vi}(\mathbf{W}^{(\ell-1)})^\top \mathbf{h}_i^{(\ell-1)}$, where we denote $\mathbf{s}_i^\top$ the $i$-th row of matrix $\mathbf{S}$ and $s_{ij}$ the $i, j$-th entry of $\mathbf{S}$. Then by chain rule, we obtain

$$\begin{aligned} \left\| \frac{\partial \mathbf{h}_v^{(\ell)}}{\partial \mathbf{x}_u} \right\| &= \left\| \mathrm{diag}\left( (\mathbf{W}^{(\ell-1)})^\top (\mathbf{H}^{(\ell-1)})^\top \mathbf{a}_v \right) \odot \left( \sum_{i_\ell=1}^n s_{vi_{\ell-1}}(\mathbf{W}^{(\ell-1)})^\top \frac{\partial \mathbf{h}_{i_{\ell-1}}^{(\ell-1)}}{\partial \mathbf{x}_u} \right) \right\| \\ &\leq \left\| \sum_{i_{\ell-1}=1}^N s_{vi_{\ell-1}}(\mathbf{W}^{(\ell-1)})^\top \frac{\partial \mathbf{h}_{i_{\ell-1}}^{(\ell-1)}}{\partial \mathbf{x}_u} \right\| \\ &\leq \left\| \sum_{i_{m-1}, i_{m-2}, \ldots, i_0} s_{vi_{m-1}} s_{i_{\ell-1}i_{\ell-2}} \cdots s_{i_1 i_0}(\mathbf{W}^{(\ell-1)})^\top (\mathbf{W}^{(\ell-2)})^\top \cdots (\mathbf{W}^{(0)})^\top \frac{\partial \mathbf{h}_{i_0}^{(0)}}{\partial \mathbf{x}_u} \right\| \\ &= \left( \sum_{i_{\ell-1}, i_{\ell-2}, \ldots, i_1} s_{vi_{\ell-1}} s_{i_{\ell-1}i_{\ell-2}} \cdots s_{i_1 u} \right) \left\| (\mathbf{W}^{(\ell-1)})^\top (\mathbf{W}^{(\ell-2)})^\top \cdots (\mathbf{W}^{(0)})^\top \right\| \\ &\leq \mathrm{w}^\ell \left( \mathbf{S} \right)_{v,u}^\ell \end{aligned}$$

where we have applied the second inequality recursively to obtain the third inequality. $\square$

## C.3 PROOF OF LEMMA 3

In this section we show the proof of the trade-off Lemma (Lemma 3) between over-smoothing and over-squashing. It is worth noting that we still set $\theta_2 = \theta_1 = \mathbf{1}_N$. Although with the same meaning as Lemma 3, below we first show the full version of Lemma 3 in the following.

**Lemma 5** (Trade-off). *For two G-MHKGs namely G-MHKG(1) and G-MHKG(2), if G-MHKG(1) has lower over-squashing than G-MHKG(2) i.e., $\mathrm{e}^{f_1(\mathbf{\Lambda})} + \mathrm{e}^{g_1(\mathbf{\Lambda})} < \mathrm{e}^{f_2(\mathbf{\Lambda})} + \mathrm{e}^{g_2(\mathbf{\Lambda})}$, where $f_1, g_1$ and $f_2, g_2$ are filtering functions of two models, respectively. Then on any layer i.e., from $\mathbf{H}^{(\ell-1)}$ to $\mathbf{H}^{(\ell)}$. Taking $\mathbf{H}^{(\ell-1)}$ as the initial input feature at layer $\ell$, we have the following inequality in terms of Dirichlet energy of two models*

$$\mathbf{E}(\mathbf{H}^{(\ell)})_1 < \mathbf{E}(\mathbf{H}^{(\ell)})_2.$$

*In words, more feature smoothing effect is induced from lower over-squashing model G-MHKG(1).*

*Proof.* First, without loss of generality, we further let all entries of the feature representation be non-negative since this can be easily achieved by applying commonly used activation function of each

layer of G-MHKG, we omit here for simplicity reason. If G-MHKG(1) is with lower over-squashing than G-MHKG(2), then, based on Lemma 2, it is sufficiently to have:

$$\left(\mathbf{U}\left(e^{f_1(\mathbf{\Lambda})} + e^{g_1(\mathbf{\Lambda})}\right)\mathbf{U}^\top\right)_{i,j} < \left(\mathbf{U}\left(e^{f_2(\mathbf{\Lambda})} + e^{g_2(\mathbf{\Lambda})}\right)\mathbf{U}^\top\right)_{i,j} \quad \forall i,j. \tag{17}$$

Let $\mathbf{\Lambda}_1^* = e^{f_1(\mathbf{\Lambda})} + e^{g_1(\mathbf{\Lambda})}$ and $\mathbf{\Lambda}_2^* = e^{f_2(\mathbf{\Lambda})} + e^{g_2(\mathbf{\Lambda})}$ be the diagonal matrices with entries of the filtered eigenvalues of two models, respectively. According to Eq. (17), we have $(\mathbf{\Lambda}_1^*)_{i,i} < (\mathbf{\Lambda}_2^*)_{i,i} \, \forall i \in [1, N]$. Since G-MHKG is linear i.e., at every iteration, the model assigns same propagation rule to the node features. For any fixed graph $\mathcal{G}$, it is easy to verify $\mathbf{E}(\mathbf{H})_1 < \mathbf{E}(\mathbf{H})_2$ according to the definition of Dirichlet energy (i.e., $\mathbf{E}(\mathbf{H}) = \mathrm{Tr}(\mathbf{H}^\top \widehat{\mathbf{L}}\mathbf{H})$). $\square$

**Remark 5.** The result in Lemma 3 is general and it is not hard to apply it to the single-scale GNNs. Furthermore, we note that the conclusion we reached is based on the sufficient requirement $(\mathbf{U}(e^{f_1(\mathbf{\Lambda})} + e^{g_1(\mathbf{\Lambda})})\mathbf{U}^\top)_{i,j} < (\mathbf{U}(e^{f_2(\mathbf{\Lambda})} + e^{g_2(\mathbf{\Lambda})})\mathbf{U}^\top)_{i,j} \, \forall i, j$ and without considering the effect of filtering matrices (i.e., $\theta_2$ and $\theta_1$). We leave the discussion and analysis for more complicated cases in future work.

## C.4 PROOF OF THEOREM 2

In this section, we show the proof of Theorem 2 which indicates the G-MHKG can handle both over-smoothing and over-squashing issues with so-called D-HFD.

**Theorem 5** (Repeat of Theorem 2). *G-MHKG is capable of handling both two issues with HFD dynamic and non-increasing over-squashing. Specifically, let $k$ be the number of connected components of $\mathcal{G}$ and $\theta_1$, $\theta_2 \geq 0$ then both two issues can be sufficiently handled by setting $\mathrm{diag}(\theta_1)_{i,i} = \mathrm{diag}(\theta_2)_{i,i} = 0 \; i \in [1, k]$ and $\mathrm{diag}(\theta_1)e^{f(\mathbf{\Lambda}_{i,i})} + \mathrm{diag}(\theta_2)e^{g(\mathbf{\Lambda}_{i,i})} < \mathbf{\Lambda}_{i,i} \; i \in [k+1, N]$.*

*Proof.* First, based on the spectral graph theory (Chung, 1997), if $\mathcal{G}$ has $k$ connected components, then $\lambda_1 = \lambda_2 = \cdots = \lambda_k = 0$. Therefore, when $\lambda = 0$, one can only let $\mathrm{diag}(\theta_1) = \mathrm{diag}(\theta_2) = 0$, otherwise the result of $\mathrm{diag}(\theta_1)e^{f(0)} + \mathrm{diag}(\theta_2)e^{g(0)}$ will be greater than 0, suggesting to an increase of over-squashing of the model according to Lemma 2. On the other hand, when $\lambda \neq 0$, as long as $\mathrm{diag}(\theta_1)e^{f(\lambda)} + \mathrm{diag}(\theta_2)e^{g(\lambda)} < \lambda$, the model is capable of decreasing over-squashing. $\square$

## C.5 PROOF OF THEOREM 3 AND EXTENSIONS

The theorem regarding to the improvised LFD is repeated as follows:

**Theorem 6** (Repeat of Theorem 3). *Suppose $\theta_1$ and $\theta_2 \geq 0$, then it is impossible for the model to be LFD and decrease OSQ. In other words, there must exist at least one $\mathbf{\Lambda}^*$ such that $\mathrm{diag}(\theta_1)e^{f(\mathbf{\Lambda}^*)} + \mathrm{diag}(\theta_2)e^{g(\mathbf{\Lambda}^*)}$ is (1) monotonically increasing (HFD) or (2) with constant quantity (thus not LFD) or (3) monotonically decreasing with image greater than $\mathbf{\Lambda}$ thus OSQ is increased.*

*Proof.* The claims of the theorem can be easily verified according to the definitions of L/HFD. Again, without lost of generality, we additionally require $\mathrm{diag}(\theta_1)e^{f(\lambda)} + \mathrm{diag}(\theta_2)e^{g(\lambda)}$ is still *monotonic* via graph structure domain. As the result of $\mathrm{diag}(\theta_1)e^{f(\lambda)} + \mathrm{diag}(\theta_2)e^{g(\lambda)} \geq 0$ to induce a decrease of OSQ, it is sufficiently to require $\mathrm{diag}(\theta_1)e^{f(\mathbf{\Lambda})} + \mathrm{diag}(\theta_2)e^{g(\mathbf{\Lambda})} = 0$ for first $k$ zero eigenvalues and $\mathrm{diag}(\theta_1)e^{f(\mathbf{\Lambda})} + \mathrm{diag}(\theta_2)e^{g(\mathbf{\Lambda})} < \lambda$ for the rest of $\lambda$. As we additionally require the model is LFD, meaning that the function $\mathrm{diag}(\theta_1)e^{f(\mathbf{\Lambda})} + \mathrm{diag}(\theta_2)e^{g(\mathbf{\Lambda})}$ is monotonically decrease. Accordingly, there must exist at least one subset of the eigenvalue in which $\mathrm{diag}(\theta_1)e^{f(\mathbf{\Lambda})} + \mathrm{diag}(\theta_2)e^{g(\mathbf{\Lambda})}$ is increasing, since otherwise the filtering results will be less than 0. Additionally, it is not difficult to verify that the increase part of $\mathrm{diag}(\theta_1)e^{f(\mathbf{\Lambda})} + \mathrm{diag}(\theta_2)e^{g(\mathbf{\Lambda})}$ is conflict with either LFD or decreasing OSQ by checking the cases listed in the Theorem, we omit it here. $\square$

We additionally show the equivalence between the situation described in Theorem 3 and GRAND++ (Thorpe et al., 2021).

**Corollary 1.** Suppose $\theta_1$ and $\theta_2 \geq 0$, if $\mathrm{diag}(\theta_1)_{i,i} = \mathrm{diag}(\theta_2)_{i,i} > 0 \; i \in [1, k]$, where $k$ is the number of connected components of $\mathcal{G}$. Further let $(\mathbf{U}\mathrm{diag}(\theta_1)e^{f(\lambda_i)} + \mathrm{diag}(\theta_2)e^{g(\lambda_i)}\mathbf{U}^\top)_{i,j} =$

$\widehat{\mathbf{L}}^*_{i,j} \leq \widehat{\mathbf{L}}_{i,j}$, $i \in [k+1, N]$. Then G-MHKG is equivalent to a special GRAND++ model with special design of source term.

*Proof.* Without loss of generality, we further let $\operatorname{diag}(\theta_1)_{i,i} = \operatorname{diag}(\theta_2)_{i,i} = c, c > 0\ i \in [1, k]$, then we have:

$$\mathbf{U}\operatorname{diag}(\theta_1)\mathrm{e}^{f(\lambda_i)} + \operatorname{diag}(\theta_2)\mathrm{e}^{g(\lambda_i)}\mathbf{U}^\top = \mathbf{U}\operatorname{diag}\begin{bmatrix} c_1\mathrm{e}^{f(0)} \\ c_2\mathrm{e}^{f(0)} \\ \vdots \\ c_k\mathrm{e}^{f(0)} \\ (\theta_1)_{k+1}\mathrm{e}^{f(\lambda_{k+1})} \\ \vdots \\ (\theta_1)_N\mathrm{e}^{f(\lambda_N)} \end{bmatrix} + \operatorname{diag}\begin{bmatrix} c_1\mathrm{e}^{g(0)} \\ c_2\mathrm{e}^{g(0)} \\ \vdots \\ c_k\mathrm{e}^{g(0)} \\ (\theta_2)_{k+1}\mathrm{e}^{g(\lambda_{k+1})} \\ \vdots \\ (\theta_2)_N\mathrm{e}^{g(\lambda_N)} \end{bmatrix}\mathrm{e}^{g(\lambda_i)}\mathbf{U}^\top,$$

(18)

where we have $c_1 = c_2 \cdots = c_k = c$. Let $\mathbf{U}\operatorname{diag}(\theta_1)\mathrm{e}^{f(\lambda_i)} + \operatorname{diag}(\theta_2)\mathrm{e}^{g(\lambda_i)}\mathbf{U}^\top\mathbf{X}\mathbf{W} = \widehat{\mathbf{L}}^*\mathbf{X}\mathbf{W}$, then Eq. (18) can be further expressed as:

$$\widehat{\mathbf{L}}^*\mathbf{X}\mathbf{W} = \mathbf{U}\operatorname{diag}\begin{bmatrix} 0 \\ 0 \\ \vdots \\ 0 \\ (\theta_1)_{k+1}\mathrm{e}^{f(\lambda_{k+1})} + (\theta_2)_{k+1}\mathrm{e}^{g(\lambda_{k+1})} \\ \vdots \\ (\theta_1)_N\mathrm{e}^{f(\lambda_N)} + (\theta_2)_{k+1}\mathrm{e}^{g(\lambda_N)} \end{bmatrix} + \operatorname{diag}\begin{bmatrix} c_1(\mathrm{e}^{f(0)} + \mathrm{e}^{g(0)}) \\ c_2(\mathrm{e}^{f(0)} + \mathrm{e}^{g(0)}) \\ \vdots \\ c_k(\mathrm{e}^{f(0)} + \mathrm{e}^{g(0)}) \\ 0 \\ \vdots \\ 0 \end{bmatrix}\mathbf{U}^\top\mathbf{X}\mathbf{W}.$$

(19)

Now we see Eq. (19) can be treated as a special diffusion process on $\widehat{\mathbf{L}}^*$ that the results from filtering function and matrices yield first $k$ eigenvalues as 0 and the rest as $\theta_1\mathrm{e}^{f(\lambda)} + \theta_2\mathrm{e}^{g(\lambda)}$. Since we have assumed $c_1 = c_2 \cdots = c_k$ and both filtering functions are not periodical, we have

$$\mathbf{U}\operatorname{diag}\begin{bmatrix} c_1(\mathrm{e}^{f(0)} + \mathrm{e}^{g(0)}) \\ c_2(\mathrm{e}^{f(0)} + \mathrm{e}^{g(0)}) \\ \vdots \\ c_k(\mathrm{e}^{f(0)} + \mathrm{e}^{g(0)}) \\ 0 \\ \vdots \\ 0 \end{bmatrix}\mathbf{U}^\top\mathbf{X},$$

(20)

as a fixed design of source term, and this completes the proof. $\square$

**Remark 6.** The conditions on $k$ for inducing delayed LFD indicate a new perspective on enhancing GNNs for fitting the non-connected graphs. Based on the property of delayed LFD, one can first adjust the quantity of $\theta_1$ and $\theta_2$ such that $\widehat{\mathbf{L}}^*_{i,j} \geq \widehat{\mathbf{L}}_{i,j}$, $i \in [1, k]$ suggesting to assign different amount of variations to different connected components. Then a LFD with decrease OSQ appears with $\widehat{\mathbf{L}}^*_{i,j} \leq \widehat{\mathbf{L}}_{i,j}$, $i \in [k+1, N]$ indicating GNN is smoothing the node features within each connected components with a decreased over-squashing. We leave more detailed exploration in the future research.

## D   EXPERIMENT DETAILS

In this section, we included detailed information on our empirical studies. Followed by the sequence of experiments included in the main page. We order this section as follows:

Table 2: Statistics of the datasets, $\mathcal{H}(\mathcal{G})$ represent the level of homophily.

| Datasets | Class | Feature | Node | Edge | $\mathcal{H}(\mathcal{G})$ |
|---|---|---|---|---|---|
| Cora | 7 | 1433 | 2708 | 5278 | 0.825 |
| Citeseer | 6 | 3703 | 3327 | 4552 | 0.717 |
| PubMed | 3 | 500 | 19717 | 44324 | 0.792 |
| Arxiv | 23 | 128 | 169343 | 1166243 | 0.681 |
| Wisconsin | 5 | 251 | 499 | 1703 | 0.150 |
| Texas | 5 | 1703 | 183 | 279 | 0.097 |
| Cornell | 5 | 1703 | 183 | 277 | 0.386 |

- We included the the details on the baseline models, summary statistics, and parameter searching space in Section D.1.

- In addition to the Section 8.2, we include the setup of inducing four types of dynamics in Section D.2.

- The computational complexity of our model is discussion in Section D.3.

## D.1 EXPERIMENT DETAILS

**Summary of baseline models**   We include the introduction of the baseline models and the reason of selecting them here.

- **MLP**: Standard feed forward multiple layer perceptron, serving as the stat-of-the-art of most of heterophily graphs.

- **GCN** (Kipf & Welling, 2016): GCN is the first of its kind to implement linear approximation to spectral graph convolutions.

- **GAT** (Veličković et al., 2018): GAT generates attention coefficient matrix that element-wisely multiplied on the graph adjacency matrix according to the node feature based attention mechanism via each layer to propagate node features via the relative importance between them.

- **GIN** (Xu et al., 2019) Graph isomorphism networks first explores the equivalence between the message passing paradigm in GNNs and the so-called Weisfeiler and Lehman isomorphism test (WL). GIN shows the upper bound of the expressive power of message passing GNNs is as same as WL test.

- **HKGCN** (Zhao et al., 2020) is the first paper to deploy the graph heat kernel to enhance GNN performance via homophily graphs.

- **GRAND** (Chamberlain et al., 2021a) linked the graph heat equation and related PDEs to the GNNs' propagation.

- **SJLR** (Giraldo et al., 2022) is the first paper considers both over-smoothing and over-squashing issues, providing so-called curvature rewiring approach to mitigate both issues.

- **UFG: Graph Framelet** (Zheng et al., 2022) a class of GNNs built upon framelet transforms utilizes framelet decomposition to effectively merge graph features into low-pass and high-pass spectra.

**Summary of included benchmarks and parameter searching space**   We first include the summary statistics of the included benchmarks in Table. 2. In regarding to the parameter searching space, we tuned hyper-parameters using grid search method. The search space for learning rate was in $\{0.1, 0.05, 0.01, 0.005\}$, number of hidden units in $\{16, 32, 64\}$, weight decay in $\{0.05, 0.01, 0.005\}$, dropout in $\{0.3, 0.5, 0.7\}$. In terms of the initial warm-up coefficient, we set $\gamma = 1.1$ for all included datasets, and it is worth noting that $\gamma$ is always applied to the domain that required to be dominated. For example, for homophily graph, we apply $\gamma = 1.1$ to $\mathrm{e}^{f(\mathbf{\Lambda})}$ whereas for heterophily graph $\gamma$ is multiplied with $\mathrm{e}^{g(\mathbf{\Lambda})}$.

## D.2 SETUP FOR FOUR TYPES OF DYNAMICS

In this section, we show how four different model dynamics can be induced by simply change the quantity of $\theta_1$ and $\theta_2$ in G-MHKG. We note that this can also be served as the experimental setup for Section 8.2. In terms of inducing LFD, we set $\zeta = \theta_1/\theta_2 = 2$, for HFD+ increasing OSQ dynamic, we set $\zeta = 0.2$. Furthermore, we remain the same setting $\zeta = 0.2$ for D-HFD + increase OSQ while merely set up first $k$ components of the filtered eigenvalues as 0. Finally, we let $\theta_1 = 0.1$ and $\theta_2 = 0.3$ (thus $\zeta = 3$) to induce D-HFD + decrease OSQ also with the filtering results less than all $\lambda$. It is worth noting that we set all $\theta$ as constant vectors.

## D.3 COMPUTATIONAL COMPLEXITY

Although the propagation of both G-MHKG and MHKG requires eigendecomposition of the graph Laplacian, empirically one may refer to the following two approaches to boost the speed of the model. First, the eigendecomposition results of the graph can be stored and read once it is done by one time, therefore in fact there is only one step of eigendecomposition in both MHKG and G-MHKG. Second, even though the filtering process is conducted on the spectral domain (i.e. Eq. (3)), empirically, one can utilize polynomial approximation to approximate the $\mathbf{U}\mathbf{\Lambda}_1\mathbf{U}^\top$ and $\mathbf{U}\mathbf{\Lambda}_2\mathbf{U}^\top$, where $\mathbf{\Lambda}_1 = \mathrm{diag}(\{\mathrm{e}^{-f(\lambda_i)}\}_{i=1}^N)$ and $\mathbf{\Lambda}_2 = \mathrm{diag}(\{\mathrm{e}^{f(\lambda_i)}\}_{i=1}^N)$, respectively and then assign the filtering matrices outside the decomposition. Thereby, the computational complexity of both of our model is equivalent to the standard linear GCN (Kipf & Welling, 2016) and graph diffusion models (Chamberlain et al., 2021a).

## D.4 ADDITIONAL ABLATION STUDY

### D.4.1 ABLATION ON $\gamma$

In this section, we first conduct additional ablation study on the warm-up coefficient $\gamma$. Recall that for MHKG, we set $f(\widehat{\mathbf{L}}) = \mathbf{U}(\mathrm{e}^{\mathbf{\Lambda}})\mathbf{U}^\top$ followed by Eq. (3). For G-MHKG, we assign one initial warm-up coefficient $\gamma > 1$ that is multiplied with $f(\widehat{\mathbf{L}})$ if the graph is homophily otherwise on $g(\widehat{\mathbf{L}})$ if a graph is heterophily. In this ablation study, we fixed all other parameters and only change the value of $\gamma$ and to evaluate G-MHKG's performance changes. The values of $\gamma$ included in this test are $0.1, 0.5, 1, 1.1$ and $2, 3, 4, 5$, where $\gamma = 1.1$ was the initial value that we applied to all experiments. Taking homophily graph test as an example, multiplying $f(\widehat{\mathbf{L}})$ with $\gamma > 1$ suggests the model focus more on the low-pass filtering results, and this setting is similar to the functionality of $\zeta$ in terms of determining the dominated dynamic in G-MHKG. The learning accuracy are included in Table 3.

Table 3: Ablation on the value of $\gamma$, the first two performances are in **bold**. $\gamma$ is multiplied onto $f(\widehat{\mathbf{L}})$ if graph is homophily and onto $g(\widehat{\mathbf{L}})$ if graph is heterophily.

| $\gamma$ values | Cora | Citeseer | Pubmed | Cornell | Texas | Wisconsin |
|---|---|---|---|---|---|---|
| $\gamma = 0.1$ | 75.0±1.1 | 60.4±1.3 | 62.8±0.9 | 83.5±0.7 | 79.9±0.1 | 81.8±0.3 |
| $\gamma = 0.5$ | 81.3±0.4 | 68.9±2.1 | 74.9±1.2 | 86.4±0.8 | 81.2±0.4 | 83.9±0.3 |
| $\gamma = 1$ | 83.0±0.7 | 72.4±0.5 | 79.4±1.1 | 89.2±0.4 | 88.4±1.5 | 87.1±0.4 |
| $\gamma = 1.1$(G-MHKG) | **83.5±0.2** | **72.8±0.2** | **80.1±0.4** | 90.2±0.9 | **89.6±0.6** | **91.2±1.5** |
| $\gamma = 2$ | 83.2±0.4 | 72.5±0.4 | **80.0±0.8** | 90.5±1.4 | 90.3±0.8 | 87.9±0.1 |
| $\gamma = 3$ | 83.1±0.8 | **72.9±0.4** | 79.4±0.3 | 91.0±0.4 | 88.7±0.7 | **91.5±1.5** |
| $\gamma = 4$ | **83.7±0.7** | 69.9±0.8 | 79.5±0.9 | 88.4±1.3 | 85.2±0.8 | 88.7±0.4 |
| $\gamma = 5$ | 80.3±0.2 | 71.3±0.2 | 78.3±0.6 | 80.5±0.9 | 82.3±0.9 | 82.9±2.0 |

**Results** Based on the results in Table 3, one can find that when $\gamma < 1$, meaning that model is not concentrate on the frequency domain that shall dominant the dynamic, the learning accuracy for both homo and heterophily graphs are relatively low. With the increase of $\gamma$, the performance of the model gradually increase, suggesting an increasing power in adapting the different types of graphs. Furthermore, we also observe that there is an certain accuracy drop when model is aligned with relatively large $\gamma$ (i.e., $\gamma = 5$). One possible interpretation of this observation is when the desirable dynamic is significantly over its counterparts, the model can be simply regarded as a single scale GNN that only shrink/sharpen the node features. In generally, this may not be desirable for both types of graphs unless they are purely homo/heterophily. Finally, it is worth noting that this ablation

study is with the same propose on testing model's sensitivity on the specific time step $t$ in (Zhao et al., 2020).

### D.4.2 ABLATION ON THE FORM OF FILTERING FUNCTIONS

In the formulation of MHKG and experimental setup, we mentioned that in general in MHKG, $f$ can take the form of any *monotonic positive* function on $\mathbf{\Lambda}$ rather than with the base of e. In this section, we conduct the ablation study on the form of $f$. First we set $f(\widehat{\mathbf{L}}) = \widehat{\mathbf{L}} = \mathbf{U}\mathbf{\Lambda}\mathbf{U}^\top$ by dropping the exponential base that initially defined in MHKG. Accordingly, the form of MHKG becomes:

$$\mathbf{H}^{(\ell)} = \mathbf{U}\mathrm{diag}(\theta_1)\mathbf{\Lambda}_1\mathbf{U}^\top\mathbf{H}^{(\ell-1)}\mathbf{W}^{(\ell-1)} + \mathbf{U}\mathrm{diag}(\theta_2)\mathbf{\Lambda}_2\mathbf{U}^\top\mathbf{H}^{(\ell-1)}\mathbf{W}^{(\ell-1)}, \qquad (21)$$

where $\mathbf{\Lambda}_1 = -f(\mathbf{\Lambda}) = -\mathbf{\Lambda}$ and $\mathbf{\Lambda}_2 = f(\mathbf{\Lambda}) = \mathbf{\Lambda}$. We name the model defined in Eq. (21) as MHKG-I standing for the initial version of MHKG.

Furthermore, one can let $f(\cdot) = \sin(\cdot)$ and $g(\cdot) = \cos(\cdot)$ and the corresponding model becomes:

$$\mathbf{H}^{(\ell)} = \mathbf{U}\mathrm{diag}(\theta_1)\sin(\mathbf{\Lambda}/8)\mathbf{U}^\top\mathbf{H}^{(\ell-1)}\mathbf{W}^{(\ell-1)} + \mathbf{U}\mathrm{diag}(\theta_2)\cos(\mathbf{\Lambda}/8)\mathbf{U}^\top\mathbf{H}^{(\ell-1)}\mathbf{W}^{(\ell-1)}, \quad (22)$$

where the inclusion of the eigenvalue scaling $\mathbf{\Lambda}/8$ is to ensure the filtering functions are monotonic in their domains, and such setting is align with the popular graph framelet model (Zheng et al., 2021). In fact, without considering the filtering matrices $\mathrm{diag}(\theta_1)$ and $\mathrm{diag}(\theta_2)$, it is not difficult to verify that $f(\cdot)^2 + g(\cdot)^2 = 1$, suggesting a perfect decomposition and reconstruction process on the graph node features, that is $\mathbf{U}\sin^2(\mathbf{\Lambda}/8) + \cos^2(\mathbf{\Lambda}/8)\mathbf{U}^\top\mathbf{H} = \mathbf{H}$. It is worth noting that under this settings of $f$ and $g$, such decomposition and reconstruction process is same as the *tightness* principle of graph framelet. Therefore, one can interpret the model defined in Eq. (22) is a special type of graph framelet without tightness. Accordingly, we name the model as G-MHKG-F, standing for generalized multi-scale heat kernel GCN align with graph framelet. We now conduct ablation studies on the form of filtering functions via MHKG-I and G-MHKG-F.

Table 4: Ablation on the form of filtering functions. Top two in **bold**.

| Methods | Cora | Citeseer | Pubmed | Cornell | Texas | Wisconsin |
|---------|------|----------|--------|---------|-------|-----------|
| MLP | 55.1 | 59.1 | 71.4 | **91.3±0.7** | **92.3±0.7** | **91.8±3.1** |
| GCN | 81.5±0.5 | 70.9±0.5 | 79.0±0.3 | 66.5±13.8 | 75.7±1.0 | 66.7±1.4 |
| GAT | 83.0±0.7 | **72.0±0.7** | 78.5±1.0 | 76.0±1.1 | 78.8±0.9 | 71.0±4.6 |
| GIN | 78.6±1.2 | 71.4±1.1 | 76.9±0.6 | 78.0±1.9 | 74.6±0.8 | 72.9±2.5 |
| HKGCN | 81.9±0.9 | 72.4±0.4 | **79.9±0.3** | 74.2±2.1 | 82.4±0.7 | 85.5±2.7 |
| GRAND | 82.9±1.4 | 70.8±1.1 | 79.2±1.5 | 72.2±3.1 | 80.2±1.5 | 86.4±2.7 |
| UFG | **83.3±0.5** | 71.0±0.6 | **79.4±0.4** | 83.2±0.3 | 82.3±0.9 | **91.9±2.1** |
| SJLR | 81.3±0.5 | 70.6±0.4 | 78.0±0.3 | 71.9±1.9 | 80.1±0.9 | 66.9±2.1 |
| MHKG-I | 80.1±1.2 | 70.1±0.6 | 71.3±0.5 | 81.4±0.3 | 77.6±0.5 | 69.2±1.9 |
| G-MHKG-F | **83.1±0.2** | **72.0±0.8** | 78.5±0.4 | **88.2±1.3** | **86.1±0.4** | 84.7±0.8 |

**Results** Based on the results included in Table 4, the performances of MHKG-I are in general, worse than many baseline models. The reason for this is because with the negative filtering result (i.e., $-\mathbf{\Lambda}_{ii} < 0$) from the high-pass domain, the matrix $\mathbf{U}\mathrm{diag}(\theta_1)\mathbf{\Lambda}_1 + \mathrm{diag}(\theta_2)\mathbf{\Lambda}_2\mathbf{U}^\top$ may no longer be a SPD matrix without controlling the quantity of $\theta$. This directly suggests the importance of incorporating the base (i.e., e) for the setting of MHKG. Furthermore, one can observe that G-MHKG-F shows nearly identical results compared to graph framelet (UFG) via homophily graph and comparable or even superior performances via heterophily graphs. The first observation might require a further exploration on whether so-called *tightness* principle is significant needed via practical graph learning tasks. The second observation suggests graph framelet and G-MHKG-F can naturally adapt to heterophily graph. We note that one can verify that both framelet and G-MHKG-F can induce both L/HFD dynamic by simply applying the proof of Theorem 1 and similar works have been done in (Han et al., 2022).

