_1(\boldsymbol{\Lambda})} + \mathrm{e}^{g_1(\boldsymbol{\Lambda})} < \mathrm{e}^{f_2(\boldsymbol{\Lambda})} + \mathrm{e}^{g_2(\boldsymbol{\Lambda})}$*, where* $f_1, g_1$ *and* $f_2, g_2$ *are filtering functions of two models, respectively. Then on any layer i.e., from* $\mathbf{H}^{(\ell-1)}$ *to* $\mathbf{H}^{(\ell)}$*. Taking* $\mathbf{H}^{(\ell-1)}$ *as the initial input feature at layer* $\ell$*, we have the following inequality in terms of Dirichlet energy of two models:* $\mathbf{E}(\mathbf{H}^{(\ell)})_1 < \mathbf{E}(\mathbf{H}^{(\ell)})_2$*. In words, more feature smoothing effect is induced from lower over-squashing model G-MHKG(1).*

We include the proof in Appendix C.3. Our conclusion can be applied to single-scale GNNs. To gain a clear understanding of the trade-off described in Lemma 3, one can consider that since the sum of the filtering functions of G-MHKG(2) over G-MHKG(1) indicating a higher Dirichlet energy of the node features, and thus G-MHKG(2) produces less smoothing than G-MHKG(1). Meanwhile, the higher values of eigenvalues from G-MHKG(2) also increase model's over-squashing based on Lemma 2 and Remark 3. These observations directly suggest that imposing more sharpening effect in a model to prevent over-smoothing will lead to more over-squashing.

## 7   TIME MANIPULATION: HOW TO HANDLE TWO ISSUES AND BEYOND

While we have illustrated the fundamental relationship (trade-off) between over-smoothing and over-squashing, whether GNN models can handle both issues naturally becomes the next question. Specifically, one shall require a GNN to be HFD to avoid over-smoothing and increasing the quantities in $\mathbf{S}$ to decrease over-squashing. In terms of G-MHKG, one can see that due to the property of exponential function, it is not possible to have the summation of the diagonal position of $\mathrm{diag}(\theta_1)\mathrm{e}^{f(\boldsymbol{\Lambda})} + \mathrm{diag}(\theta_2)\mathrm{e}^{g(\boldsymbol{\Lambda})} = 0$ unless we set $\theta_1 = \theta_2 = 0$ under the situation that $\mathrm{e}^{f(\boldsymbol{\Lambda})} \neq \mathrm{e}^{g(\boldsymbol{\Lambda})}$, or $\theta_1 = -\theta_2$ when $\mathrm{e}^{f(\boldsymbol{\Lambda})} = \mathrm{e}^{g(\boldsymbol{\Lambda})}$. Nonetheless, to satisfy the condition, we delay the occurrence of HFD by setting $\theta_1 = \theta_2 = 0$ when $\lambda = 0$, and our result is summarized in the following theorem.

**Theorem 2** (Delayed HFD (D-HFD)). *G-MHKG is capable of handling both two issues with HFD dynamic and non-increasing over-squashing. Specifically, let* $k$ *be the number of connected compo-*

*nents of $\mathcal{G}$ and $\theta_1$, $\theta_2 \geq 0$ then both two issues can be sufficiently handled by setting* $\operatorname{diag}(\theta_1)_{i,i} = \operatorname{diag}(\theta_2)_{i,i} = 0$ $i \in [1, k]$ *and* $\operatorname{diag}(\theta_1)\mathrm{e}^{f(\boldsymbol{\Lambda}_{i,i})} + \operatorname{diag}(\theta_2)\mathrm{e}^{g(\boldsymbol{\Lambda}_{i,i})} < \boldsymbol{\Lambda}_{i,i}$ $i \in [k+1, N]$.

We included the detailed proof in Appendix C.4 with additional discussions and clarifications. Importantly from Theorem 2, the reason why we consider zero eigenvalues of the graph is to sufficiently decrease the over-squashing, one shall require the filtered eigenvalues are not larger than the graph spectra. Therefore when eigenvalues are 0, the sufficient condition is to require the filtering results equal to 0 to maintain the over-squashing level. Figure 2 shows the comparison between different kinds of filtering functions in regard to the effect on over-smoothing and over-squashing. In addition, although Theorem 2 show how a D-HFD model handle two issues, this conclusion is more applicable to heterophily graphs according to Remark 1. However, as we will show in the next theorem, it is not possible for the model to be LFD and decrease over-squashing.

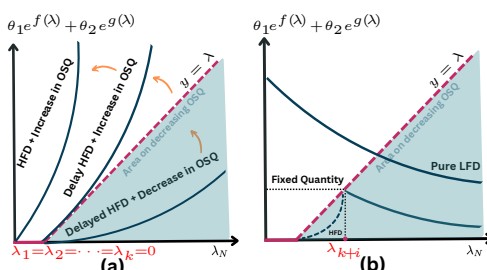

Figure 2: The figure on the left represents different types of HFD filtering outcomes and the trade-off between two issues. One can check that to induce more sharpening (filtering function from bottom to top), the model will suffer more from OSQ. The figure on the right illustrates the situation described in Theorem 3.

**Theorem 3.** *Suppose $\theta_1$ and $\theta_2 \geq 0$, then it is impossible for the model to be LFD and decrease OSQ. In other words, there must exist at least one $\boldsymbol{\Lambda}^*$ such that* $\operatorname{diag}(\theta_1)\mathrm{e}^{f(\boldsymbol{\Lambda}^*)} + \operatorname{diag}(\theta_2)\mathrm{e}^{g(\boldsymbol{\