# OpenReview forum: "Unifying over-smoothing and over-squashing in graph neural networks: A physics informed approach and beyond"
_ICLR.cc/2024/Conference — ICLR 2024 Conference Withdrawn Submission_

### Official Review · Reviewer_C4jf · 2023-10-30

**Soundness:** 2 fair
**Presentation:** 2 fair
**Contribution:** 2 fair
**Rating:** 3
**Confidence:** 4

**Summary:**

Graph Neural Networks (GNNs) are powerful tools for processing graph-structured data, but they face challenges like over-smoothing, over-squashing, and limited expressive power. This study introduces a new approach, drawing inspiration from the time-reversal principle in physics, which inverts the direction of the graph heat equation. This inversion leads to the development of the Multi-Scale Heat Kernel GCN (MHKG) and its generalized version, G-MHKG. These models address the aforementioned challenges by balancing the smoothing and sharpening effects on node features. The research uncovers a trade-off between over-smoothing and over-squashing. Empirical results aim to show the effectiveness of the proposed models.

**Strengths:**

* The motivation of this study is nicely introduced.
* Balancing smoothing and non-smoothing terms in a GNN is very intuitive (although it has to be mentioned that is not the first work doing that).
* Reverting the graph heat equation to address over-smoothing is an interesting and intuitive idea.

**Weaknesses:**

Section 5 is fully taken from Di Giovanni et al., 2023, and is not just "similar" or concurrent. Therefore, the authors have to explicitly state that the notion of maximal mixing is taken from Di Giovanni et al., 2023, as well as state that the notion of OSQ is copied/lifted from Di Giovanni et al., 2023, Appendix E. In its current form it suggests that the authors present something similar/concurrent to Di Giovanni et al., 2023, when in fact it is fully taken from there.


The provided experimental section is very problematic:
1. As I understand, Figure 4 should serve as evidence of correlation between different model set-ups and performance. It, however, mostly shows that no deep configuration of the proposed methods work, neither on homophilic datasets nor on heterophilic datasets, as the performance seems to drastically drop for increasing number of layers. This raises the question if the proposed model in fact solves over-smoothing (or over-squashing), as it was claimed.

2. The considered configurations of MHKG and G-MHKG in 8.3 are apparently GNNs with only two layers. I presume this was done because deeper models do not work. However, this contradicts the whole claims before, as over-smoothing and over-squashing is only an issue for deep GNNs.

3. The paper states that the provided results in table 1 are based on the public splits of Pei et al., 2020. However, based on these splits there are much better results reported in the literature (https://arxiv.org/abs/2202.04579) for the considered baselines. For instance, GCN reaches 86.98\% test accuracy on Cora instead of the reported 81.5\%, which is significantly better than the best reported results for the model proposed in this paper (i.e., 83.5\%). This suggests that the proposed models do not perform well on real-world data.

4. It is no clear if over-squashing is an issue for any of the considered datasets. It would be interesting to test the proposed models on the synthetic ZINC experiment suggested in Di Giovanni et al., 2023, for different configurations of the model (i.e., with increasing/decreasing osq). This is an experiment explicitly designed to test osq.

**Questions:**

See weaknesses

---

### Official Review · Reviewer_YhrV · 2023-10-31

**Soundness:** 2 fair
**Presentation:** 1 poor
**Contribution:** 3 good
**Rating:** 3
**Confidence:** 4

**Summary:**

In this paper, the authors introduce G-MHKG, a modified version of the spectral GNN that takes into account both the graph heat equation and its inverse.  A trade-off between oversmoothing and oversquashing issues is shown for G-MHKG. Besides, a simple fix is proposed for G-MHKG so that it can handle both issues at the same time. When tested on diverse graph types, both heterophic and homophic, G-MHKG consistently performed well, often surpassing or matching the performance of other baseline methods.

**Strengths:**

- The introduction of a time-reversed graph heat kernel offers a fresh and captivating perspective. This approach seems to be a robust enhancement to the foundational spectral GNN framework.
- The authors provide a comprehensive theoretical analysis, addressing both oversmoothing and oversquashing challenges associated with the updated architecture.

**Weaknesses:**

The clarity of the paper is notably lacking and significantly weakens the manuscript's overall strength. Several results in the main content lack clear articulation and interpretation. Additionally, upon reviewing the appendix, I observed numerous proofs that appear to have been presented without due diligence, evidenced by omissions, typographical errors, and inaccuracies. I've outlined specific concerns in the questions that follow.

**Questions:**

- When introducing G-MHKG, a choice of dynamics is introduced in equation (4). However, this is very confusing as there are two dynamics in the equation. Do you want to consider the sum of $f(\hat{L})$ and $g(\hat{L})$ to combine them together?
  - After reading subsequent sections, I realized that $f$ is supposed to be increasing whereas $g$ is supposed to be decreasing. This should be clearly stated when considering the dynamics in equation (4).
  - Why is the non-linearity functions not involved in defining G-MHKG?
- The introduction of Definition 2 and OSQ is quite confusing as the notations are never used in later analysis or proofs. I think the authors should carefully incorporate these notations into the text or should otherwise remove them.
- Theorem 1:
  - In (14), what is the role of $\tau$? Isn't $\tau=1$?
  - In the equation below (14) starting with $vec(H(m\tau))$, I believe certain norm or absolute value signs should be included. Otherwise, I don't think the inequality holds given that $\lambda_k^W$ or $c_{k,i}(0)$ can be negative. Besides, it is not clear to me the meaning of $\leq$ between two vectors.
  - I think a proper explanation for why $(I_c\otimes \hat{L})h_\infty=\rho_{\hat{L}}h_\infty$ is needed.
- Lemma 2: am I correct that $w$ is an upper bound for $\|W^{(l)}\|$ for all $l$? This is what I infer from the proof and if so, please state this clearly in the lemma.
  - I'm not sure about the definitions of $\hat{A}^l$ and $\hat{A}^h$. It seems to me that $\hat{A}^l$ should be $U\Lambda_1U^T$ instead of $I-U\Lambda_1U^T$ by reading the proof of Lemma 2. In equation (16), a new notation $\hat{L}^*$ is introduced which I think should be the same as $S$. Otherwise, I don't understand how the formula $H^{(l)}=SH^{(l-1)}W^{(l-1)}$ is derived in the paragraph that follows equation (16).
  - The authors explained the notation $\hat{L}^l$ and $\hat{L}^h$ below (16) but these notations don't appear in (16).
  - Can the authors explain the first equality in the equation below equation (16)? What is $\odot$ and how is chain rule applied? I feel that the whole term on the left of $\odot$ is unnecessary.
- Remark 3: the authors bring up an interesting inequality $(U\cdots U^T)_{i,j}<\hat{L}){i,j}$ to quantify oversquashing. I guess this may follow from the hope that $S_{i,j}<1$ but I don't see how $(U\cdots U^T)_{i,j}<\hat{L}){i,j}$ is derived.
- Lemma 3: it seems that the authors are using a new definition for oversquashing here through "$e^{f_1(\Lambda)}+e^{g_1(\Lambda)}$". This obviously is related to remark 3. However, even if remark 3 is carefully explained, $e^{f_1(\Lambda)}+e^{g_1(\Lambda)}$ only corresponds to the upper bound for the oversquashing defined in this paper instead of the oversquashing itself. So either the authors need to redefine the oversquashing or they should change the statement in lemma 3.
  - Is equation (17) a condition inherent to the statement? Or is it something that needs to be proved? Later the authors derived $\Lambda_1^*<\Lambda_2^*$ which confused me even more: isn't this the assumption in the statement?
  - Given (17), I think there is no need for the nonnegativity assumption to guarantee that $E(H_1)<E(H_2)$.
- Theorem 2: this theorem is seemingly contradictory to Lemma 3 and I don't think the authors has done a good job to explain this. In particular, setting $\theta=0$ for 0 eigenvalues seems like a trivial choice and it is not clear immediately why it resolves the trade-off in Lemma 3. Furthermore, the statement in Theorem 2 is quite unclear by saying "... is capable of handling both..." and "... can be sufficiently handled...". What does "sufficiently handled" mean?
- Theorem 3: In the proof, the following sentences are quite confusing: ".. the function $diag(\theta_1)e^{f(\Lambda)}+..$ is monotonically decrease. Accordingly, there must exist at least one subset ... is increasing". How is the increasing property of the subset related to the decreasing monotonicity of the function?

---

### Official Review · Reviewer_JgmJ · 2023-10-31

**Soundness:** 2 fair
**Presentation:** 2 fair
**Contribution:** 3 good
**Rating:** 3
**Confidence:** 3

**Summary:**

The authors propose a multi-scale heat kernel GCN (MHKG) and analyse the impact of different choices of filtering matrices and spectral transformations on high pass and low pass filtering of the induced GNN model. They discuss how their results could impact over-squashing and over-smoothing simultaneously and conclude that time reversal enables the mixing of smoothing and sharpening effects. Furthermore, they showcase in experiments on standard homophilic and heterophilic benchmark data that different filter matrices impact the generalization performance of the induced GNN models.

**Strengths:**

- The impact of different choices of spectral filters is discussed and how they could contribute to over-smoothing or over-squashing for infinitely deep spectral graph neural networks.
- The authors propose a heat kernel that incorporates time reversal to allow for mixing smoothing and sharpening effects.
- They support their insights with theoretical findings.
- Figures 2 and 3 demonstrate that different choices of filtering matrices impact the test accuracy considerably.

**Weaknesses:**

- The theory and analogy to time reversal of the heat equation only apply to linear GNNs. The theory also discusses only infinitely deep GNNs (by taking the continuous time proxy).
- Potential lack of novelty: Is the time reversal idea not also indirectly proposed by "Understanding convolution on graphs via energies", TMLR 2023?
- The authors equate over-squashing and over-smoothing with the high pass/low pass filtering. Yet, the right amount of smoothing (or squashing) is determined by the learning task and can also be controlled by the depth of the GNN, which are not considered.
- $f$ and $g$ are not learnable. Yet, Theorem 1 and Lemma 3 suggest that the trade-off between over-squashing and over-smoothing would be resolved with the right choice of $f$ and $g$. The experiments suggest that this is supposed to be a hyper-parameter tuning decision based on knowledge of the data (e.g, if it is heterophilic or homophilic). This is computationally expensive and disregards the effect of learning $W$.
- Lemma 2 does not really give insights into the expressiveness or over-squashing phenomenon, as these would depend on the precise choice of $W$ and it affects the generalization performance (and thus the learning task). The stated inequality just follows trivially from the definition of the feature update $H$.
- Filtering matrices do not control the model dynamics and over-smoothing, as claimed by the authors. They are not even learned. Note that $W$ could compensate for many problematic choices by setting
$W^{(l)} = (H^{(l-1)})^{-1} U (\text{diag}(\theta_1)\Lambda_1 + \text{diag}(\theta_2)\Lambda_2 )^{-1}  U^T H^{(l)}_t$, where $H^{(l)}_t$ denotes the target features that would be optimal for solving a task of interest.
- My biggest concern: The proof of Theorem 1 and Lemma 3 are not convincing. Note that the Dirichlet energy also depends on the weight matrices $W$. Yet, their learning dynamics are not considered. Furthermore, it seems to be assumed that $W^l$ are identical in the compared cases. Yet, they should differ during training.
- Experiments are only conducted for linear $f$ and $g$.
- The filter matrices are tuning parameters in experiments, which makes the proposal computationally expensive.
- Experiments do not seem to lead to significant gains with respect to baseline models.

Minor points:
- The writing could benefit from a grammar and language check. Sometimes it becomes difficult to deduce the meaning of sentences.
- The adjective of homophily is homophilic and of heterophily is heterophilic.
- The notation is inconsistent. Should $f(L)$ be defined as $U f(\Lambda) U^T$ or $U \exp(f(\Lambda)) U^T$?

**Questions:**

- Which parameters are trained exactly? Only $W$? Why are the function $f$ and the filter matrices $\theta$ not trainable but tuning parameters?
If $f$ is not trainable, what motivates its choice in practice (apart from the homophily or heterophily of a task)? The authors only discuss the linear case and different filter matrices.
- Is the choice of $f$ and $g$ and the filter matrices really that detrimental from a theoretical point of view or could they also be compensated by different values of $W$?

---

### Official Review · Reviewer_xBwc · 2023-11-03

**Soundness:** 3 good
**Presentation:** 2 fair
**Contribution:** 3 good
**Rating:** 6
**Confidence:** 3

**Summary:**

This paper presents a new filtering function for spectral graph neural networks. The paper then presents some theory about the new architecture connecting to the over smoothing and over squashing. Finally the paper presents experiments backing up the theory and then shows that their model does well on real data.

**Strengths:**

I think the new idea is simple and well founded. This makes it useful.

The theory connections for the architecture is also nice. It highlight prior trends that over squashing and over smoothing need to be balanced and presents a viable path to doing so.

The paper presents some interesting experiments. Such as Figure 3.

The downstream experiments are thorough, comparing against a variety of different models. Including an MLP which I appreciate.

**Weaknesses:**

The main weakness for me would be the writing of the paper. In particular the paper presents many different formations/generalizations/specific instantiations of their framework and it difficult to keep track to model being discussed. The paper also mentions some intuition that seems to be better connected with the theory. Finally in experiments also need more details about the model. Specially, section 8.1

**Questions:**

-